# DynaEval: A Dynamic Interaction-based Evaluation Framework for Assessing LLMs in Real-world Scenarios

## Abstract

Large language models (LLMs) have shown significant advancements in diverse real-world applications, underscoring the necessity for comprehensive evaluation methodologies. Existing research about LLM evaluation usually concentrates on supervised signal-based evaluation benchmarks on domain-specific tasks, which utilize static labeled datasets to evaluate the abilities of LLMs. However, these methods often fall short in evaluating LLMs in dynamic real-world scenarios, which can be viewed as goal-driven multi-agent scenarios. In these scenarios, agents have to repeatedly obtain feedbacks and improve their outputs through cooperative or adversarial interactions in order to gradually reach their goals. To address this problem, inspired by game theory, we propose a novel dynamic interaction-based LLM evaluation framework (DynaEval) for evaluating abilities of LLMs in dynamic real-world scenarios. Specifically, we first standardize the definition of the interaction process in dynamic real-world scenarios. Next, we prove that interaction processes in evaluation tasks are equivalent to a class of dynamic games in game theory, which is beneficial to the fairness and stability of evaluation. Inspired by game theory, we propose the message pool and LLM-based referee components of DynaEval, leveraging the properties of dynamic games to ensure fairness and stability throughout the interaction and evaluation process. Moreover, we propose the synchronous interaction algorithm, which is suitable for all kinds of interactions in real-world tasks. Finally, we demonstrate the effectiveness of DynaEval through extensive experiments across four interaction-based evaluation tasks stemming from real-world scenarios. Our source code is available at `https://anonymous.4open.science/r/DynaEval-112F`.

## 1 Introduction

The rapid development of Large Language Models (LLMs) has catalyzed their incorporation into a wide array of real-world applications, such as machine translation (Lyu et al., 2023) and code generation (Zheng et al., 2023). This progress in LLM-based applications has increased the necessity for comprehensive LLM evaluations. Given the immense scale and limited interpretability of LLMs, the primary focus of these evaluations centers on assessing their proficiency in domain-specific tasks. Ultimately, the evaluation outcomes of LLMs serve as valuable guidance for users in selecting the most suitable LLMs to meet their specific requirements.

In the literature, LLM evaluation methods traditionally fall into two categories: human-based and supervised signal-based. Human-based methods (Thorleiksdóttir et al., 2022; Nguyen, 2018) involve human interrogators engaging with LLMs, with the evaluation result depending on human judgments. For instance, the Turing Test (Shieber, 2006) entails a human interrogator interacting with two anonymous participants (one being an LLM and the other a human) and tasked with distinguishing between them within a limited timeframe. In spite of their flexibility, human-based evaluation methods suffer from heavy labors and huge time costs for large-scale LLM assessments across diverse tasks. Therefore, recent research about LLM evaluation mainly concentrates on supervised signal-based evaluation benchmarks. In supervised signal-based evaluation methods (e.g., Chang et al. (2023a); Maruf et al. (2022); Zhu et al. (2023)), LLMs are tasked with producing accurate

outputs given dataset inputs. These methods offer greater automation compared to human-based evaluations, as evaluation metrics can be automatically computed by comparing dataset labels to LLM outputs. Consequently, supervised signal-based approaches have found extensive use in large-scale LLM assessments. For instance, MMLU (Hendrycks et al., 2021b) provides a supervised signal-based standardized benchmark to evaluate the performance of text models in multiple tasks.

Despite their utility and efficiency, supervised signal-based evaluation methods struggle to assess the performance of LLMs in real-world scenarios characterized by dynamic interactions and diverse roles. Specifically, in real-world scenarios, users dynamically interact with LLMs to implement their requirements (e.g., generate logically-correct and well-styled Python codes), during which LLMs repeatedly get feedback and optimize their output to gruadually meet users' requirements. This is essentially equivalent to a goal-driven multi-agent scenario (Xi et al., 2023a; Bang et al., 2023a; Xi et al., 2023b), where agents (users-based and LLMs-based) propose and optimize solutions through cooperative or adversarial interaction (Mandi et al., 2023; Fu et al., 2023) to reach their goals. In these scenarios, the dynamic interaction environment (e.g., interaction context) highly affect the performance of LLM-based agents. Therefore, LLMs need strong dynamic interaction abilities to meet users' requirements in these scenarios. However, abilities of LLMs in dynamic interactions are hard to evaluate through conventional supervised signal-based evaluation techniques because of the static evaluation environment. This limitation stems from the inherent inadequacy of static datasets used in supervised signal-based approaches to capture the complexity of dynamic real-world situations. Although some evaluation methods support multi-round conversations (e.g., Chan et al. (2023)), they still fall short in providing from the dynamic interaction environment in multi-agent scenarios, rendering them inadequate for evaluating LLMs in dynamic real-world scenarios.

To this end, we propose a novel dynamic interaction-based LLM evaluation framework (DynaEval). Inspired by research in LLM-based multi-agent systems (e.g., Hong et al. (2023); Xiong et al. (2023)), we find that it is feasible to simulate dynamic interactions in real-world multi-agent scenarios by dynamic interactions between LLM-based agents, where the variety of LLM output provides the dynamic interaction environment. Moreover, we prove that the interaction process of LLMs in these scenarios essentially belong to a class of dynamic games in game theory (Kreps & Wilson, 1982; Balbus et al., 2018), which is beneficial to the fairness and stablility of evaluation. To implement such an LLM evalution framework, we first establish the prior fairness and stability condition of DynaEval based on game theory. Next, we propose the message pool, the LLM-based referee, and the synchronous interaction algorithm for DynaEval. These components ensure the fairness and stability of evaluation by keeping the consistency between the dynamic interaction process and dynamic games in game theory. In the experiment, we implement four real scenario-based LLM evaluation tasks in DynaEval, and show that DynaEval can effectively evaluate abilities of LLMs in dynamic real-world scenarios and reveal interaction characteristics of LLMs.

## 2 METHODOLOGY

### 2.1 PRELIMINARIES

In this part, we introduce necessary mathematical notations and the goal for dynamic interaction-based LLM evaluation. To begin with, let $N$ denote the number of LLMs to be evaluated. Let $\mathcal{P} = \{P_1, P_2, \ldots, P_N\}$ denote the set of LLMs where each element denotes an LLM. Let $R$ denote an interaction-based evaluation task. For example, in the Code Generation and Review (Code G&R) task (Guo et al., 2023), programmers and reviewers listen and answer alternatively in order to generate logically correct and well-styled codes. Furthermore, let $A$ denote the set interaction histories (e.g., records of dialogues and codes in Code G&R). Let $\Theta = (\theta_1, \theta_2, \ldots, \theta_N)$ denote abilities of LLMs, where $\theta_i$ denotes the ability of $P_i \in \mathcal{P}$. Then our goal is to evaluate LLMs' abilities $\Theta$ from observed interaction histories $A$.

### 2.2 FROM INTERACTIONS TO DYNAMIC GAMES

The first challenge of DynaEval lies in the variety of interaction in real-world scenarios, which makes it difficult to uniformly model the interaction process in various tasks. Therefore, we aim to extract the common ground of various interaction process and summarize as a uniform definition. We notice that any goal-driven multi-agent interaction in real-world scenarios consists of an

*interaction goal* and an *interaction rule*. The former depicts "why to interact", such as "producing logically correct and well-styled codes" in Code G&R. The latter depicts "what and how to interact", such as "participants should generate codes and comments alternatively". Based on these findings, interaction process can be uniformly decomposed to four cyclical steps, as defined in the following.

**Definition 1.** *Interaction process of DynaEval. Given a history set $A = \emptyset$ and an interaction-based evaluation task $R = (G, C)$, where $G$ denotes the interaction goal and $C$ denotes the interaction rule, the interaction process is defined as the combination of four cyclical steps as follows:*

1. ***Selection.*** *Select a subset of LLMs $P^* \subseteq \mathcal{P}$ that can interact in the current interaction environment, i.e., receive messages and generate interaction outputs, according to $C$.*

2. ***Interaction.*** *Each LLM $P_i \in P^*$ receives messages from $A$ and generates interaction output $s_i = (P_i, w_i, t^{(env)})$ according to $G$. Here $w_i$ denotes the output content and $t^{(env)}$ denotes the interaction environment identifier, such as the index of the round.*

3. ***Recording.*** *Interactions are recorded to the history set, i.e., $A \leftarrow A \bigcup \{s_i | P_i \in P^*\}$.*

4. ***Circulation.*** *If the interaction process comes to an end according to $C$, then score the performance of LLMs according to $A$. Otherwise, the interaction process repeats.*

For instance, in Code G&R, programmers can interact in odd-numbered rounds (index starts from 1), while reviewers can interact in even-numbered rounds. Each time a programmer or reviewer interact, its interaction output is recorded by the history set. This process repeats until the programmer and the reviewer reach a consensus or the number of rounds exceeds the limitation.

The second challenge lies in the fairness and stability of evaluation. The fairness of LLM evaluation connotes that evaluation results $\Theta$ are only determined by LLMs' true abilities. However, interaction-based evaluation results can be easily affected by non-ability factors such as the information asymmetry in multi-agent interaction. On the other hand, we hope to obtain stable evaluation of LLMs. Paradoxically, dynamic interaction-based evaluation results are instable due to the variety of the observed interaction history. In pursuit of overcoming this challenge, we notice that the interaction process of DynaEval is theoretically in alignment with **extensive games with perfect information** (EGPI) in dynamic game theory (Kreps & Wilson, 1982; Apt & Simon, 2021). EGPI is a class of dynamic games where participants can dynamically take actions based on the current game environment in order to maximize their own game score (namely "payoff" in game theory). Game scores directly reflect abilities of LLMs. A formal proposition is given below. The corresponding proof is available at Appendix A.2.

**Proposition 1.** *The relationship of EGPI and interaction process of DynaEval. Let $\mathcal{D}$ denote the set of all possible interaction process of DynaEval. Let $\mathcal{E}$ denotes the set of all EGPI. Then any interaction process of DynaEval also belongs to EGPI, i.e., $\mathcal{D} \subseteq \mathcal{E}$.*

By introducing EGPI from game theory, we can overcome the fairness and stability challenge using game theory. Specifically, the fairness of the interaction process of LLMs can be ensured by the inherent *anonymity* and *perfect information* in EGPI. In EGPI, anonymity means that the real identity of participants are invisible. The anonymity can prevent unfairness from two aspects. First, it *prevents targeted adversarial interaction policies to agents with known identities*. Second, in evaluation tasks that depend on referee's rating to obtain game scores, the anonymity can effectively *prevent biased rating of participants caused by revealing identities to the referee*. On the other hand, perfect information in EGPI connotes that every participant has the equal chance to collect information and make decisions. The perfect information can be transformed to the synchronicity of interaction in multi-agent scenarios, which ensures fairness from the aspect of information asymmetry. To reach this goal, it is necessary to regularize the synchronicity of the interaction process of dynamic interaction-based evaluation tasks. To this end, we propose the following fairness condition for DynaEval:

**Condition 1.** *Fairness condition. To ensure the fairness condition of DynaEval, 1) all participant LLMs in the interaction process should be anonymous. 2) The delivery of LLMs' messages should be synchronous, i.e., in the selection phase of interaction process, all select LLMs $P_i \in P^*$ should synchronously receive messages and generate outputs according to the interaction rule $C$.*

As for the stability issue, the stability of evaluation results of LLMs can be statistically ensured by modeling the game score of LLMs in EGPI. Indeed, given the fairness condition, the game scores

directly reflect the ability of LLMs. Therefore, the term $\Theta$ can also represent game scores of LLMs. We next model $\Theta$ from the aspect of statistical distribution and illustrate the cause and solution of the stability issue. Specifically, let $\Theta_i \sim Pr(P_i)$ denote the random variable form of the ability of LLM $P_i \in \mathcal{P}$, with the probability density function (pdf) represented by $f_i^{(pdf)} : \mathbb{R} \to \mathbb{R}$. Our goal is to obtain the expectation of $\Theta_i$ (average game score) as the evaluation result, i.e.,

$$\theta_i = E[\Theta_i] = \int \theta \cdot f_i^{(pdf)}(\theta)d\theta. \tag{1}$$

Unfortunately, we cannot directly get sampling results of the *full* distribution of $\Theta_i$ from the interaction process to calculate $\theta_i$. The reason is that the performance of LLMs *depends on the interaction history*, i.e., we can only get sampling results of the *conditional* distribution $\Theta_i|A \sim Pr(P_i|A)$ from the current interaction environment. Essentially, the stability issue stems from the variety of the interaction history $A$ (the interaction environment), which is common in goal-driven multi-agent scenarios. However, we notice that by expanding the pdf of $\Theta_i$ with the law of total probability, the full distribution can be obtained through *multiple independent sampling*:

$$f_i^{(pdf)}(\theta) = \sum_A f_i^{(pdf)}(\theta|A)p_T(A), \tag{2}$$

where $p_T(\cdot)$ denotes the pdf of $A$. Although all pdfs in equation 2 are intractable, we can still estimate $\theta_i$ from game scores obtained by multiple independent running of the interaction process. That is because each independent running of the interaction process indeed samples a history set from $p_T(\cdot)$ and the corresponding conditional game score from $f_i^{(pdf)}(\cdot|A)$. Therefore, the average game score of an LLM obtained from multiple independent running of the interaction process is the consistent estimation of the expectation of $\Theta_i$, thus is the evaluation result of the LLM. In conclusion, we summarize these requirements as the stability condition of DynaEval:

**Condition 2.** *Stability condition. The dynamic interaction-based evaluation process should be run independently for multiple times until evaluation results of LLMs converge in distribution. Then the expectation of game scores are evaluation results of LLMs.*

### 2.3 The Structure of DynaEval

The dynamic interaction-based LLM evaluation framework (DynaEval) aims to implement the interaction and evaluation process in dynamic interaction-based evaluation tasks. The structure of DynaEval is shown in Figure 1. To meet the fairness and the stability condition in dynamic interaction-based evaluation tasks, targeted components and mechanisms are utilized in DynaEval. For the fairness condition, we propose the *synchronous interaction algorithm*, which utilizes the *referee* and the *message pool* to ensure the anonymity and synchronicity of the interaction process. For the stability condition, as analysed above, DynaEval utilizes *multiple independent running* of evaluation tasks to ensure the stability of evaluation results. In this part, we first introduce the two indispensable components of DynaEval, i.e., the referee and the message pool. Next, we describe in detail the synchronous interaction algorithm and illustrate how it implements the interaction process.

**Referee**. The referee in DynaEval is the supervisor of the interaction process and the judge of LLMs. In DynaEval, the referee is responsible for the *selection*, *recording*, and *circulation* in the interaction process. Specifically, in the selection phase, the referee selects the next LLM to interact based on the interaction rule $C$ defined in the interaction-based evaluation task $R = (G, C)$. In the recording phase, the referee standardizes the format of LLM outputs to avoid unfairness stemming from format bias. During the circulation phase, the referee determines whether the task has reached its termination and is responsible for evaluating the ability of the LLMs. Specifically, in evaluation tasks where obtaining rule-based scores is difficult (such as assessing code style quality), the referee generates model-based evaluation scores for these anonymous LLMs based on the task rule.

**Message Pool**. The message pool in DynaEval is the channel of the interaction of LLMs and the container of the interaction histories in the interaction process. In DynaEval, the message pool is vital for the *interaction* and the *recording* of the interaction process. Specifically, in the interaction phase, messages are read from the message pool by the referee and sent to the selected LLM. Next, in the recording phase, the message pool receives and writes the output generated by the LLM. This

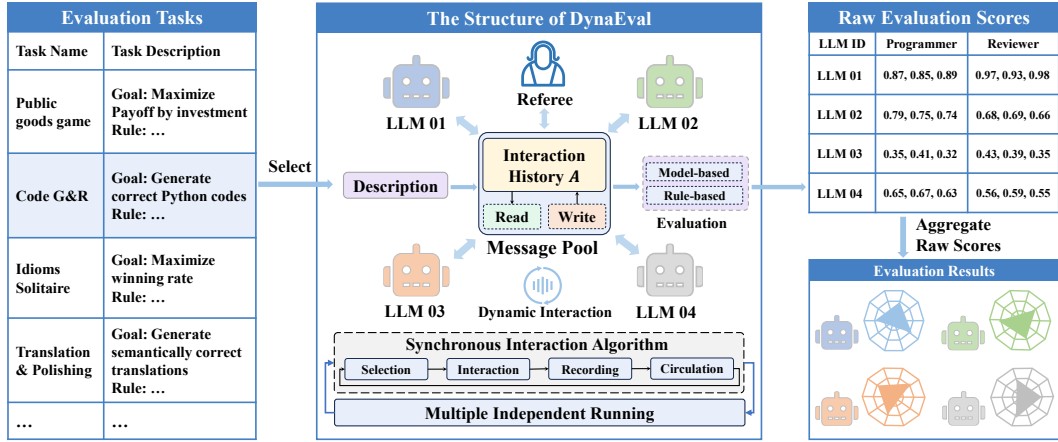

Figure 1: The dynamic interaction-based LLM evaluation framework. To evaluate LLMs, DynaEval starts with selecting a dynamic interaction-based evaluation task and input its text description and candidate LLMs to the synchronous interaction algorithm. Next, raw evaluation scores are collected from the multiple independent running of the algorithm. Finally, raw scores are aggregated to obtain LLMs' evaluation results.

action is indeed equivalent to the recording of the interaction history. A detailed illustration of the running of the message pool is presented in the synchronous interaction algorithm.

**Synchronous interaction algorithm**. The interaction process of LLMs is the core of DynaEval. In the interaction process, the *synchronicity* is fundamental to the fairness of evaluation. To achieve this, the synchronous interaction algorithm utilizes the message pool to decompose interactions into "interaction in rounds" to meet the synchronicity, and utilizes the referee to implement the complex interaction rule defined in the evaluation task.

Initially, the referee transmits the evaluation task rule to LLMs via the message pool. In the following running of the interaction process, each interaction round encompasses two phases: the receiving phase and the sending phase. In the receiving phase, which is equivalent to *selection* phase in the interaction process, LLMs retrieve messages selected by the referee based on the task rule. In the sending phase, which is equivalent to *interaction* and *recording* phase in the interaction process, LLMs output their own messages (interaction) and dispatch them to the message pool (recording). After each round, the referee assesses whether the task has concluded. If it has, a termination signal is sent to LLMs, bringing the task to a close. This step is equivalent to the *circulation* phase in the interaction process. Finally, the referee evaluates LLMs' performance and produces the results. For the pseudo-code, please refer to Appendix A.3.

## 2.4 IMPLEMENTATIONS OF EVALUATION TASKS

As a *general* LLM evaluation framework, DynaEval supports flexible design of evaluation tasks and can be adapted to a host of real-world evaluation scenarios. Without loss of generality, we propose four elaborately-designed evaluation tasks stemming from real-world scenarios to show the feasibility of DynaEval. An overview of these tasks is shown in Figure 2. For more detail about how to design evaluation tasks, please refer to Appendix A.4.

**Public Goods Game**. Public goods game (PGG) (Semmann et al., 2003; Dhami et al., 2019) is a symmetric evaluation task (i.e., all participants act the same role that has the same action set and the same goal) that requires the decision-making ability of LLMs. Specifically, at the start of a PGG, each of $N$ LLMs have the same amount of goods (e.g., dollar). In each round, all LLMs can decide whether to invest (part of or all of) its goods to the public goods pool or not. Then all invested goods will be summed and doubled by a constant factor. Then result goods are shared equally by all LLMs. For example, if two of three LLMs invested 100 dollars in total and the constant factor is $\alpha = 1.2$, then the invested goods are doubled to $100 \times 1.2 = 120$ dollars, and every LLM will get $120/4 = 30$ dollars, including those who did not invest. The payoff function of each LLM is the total amount of

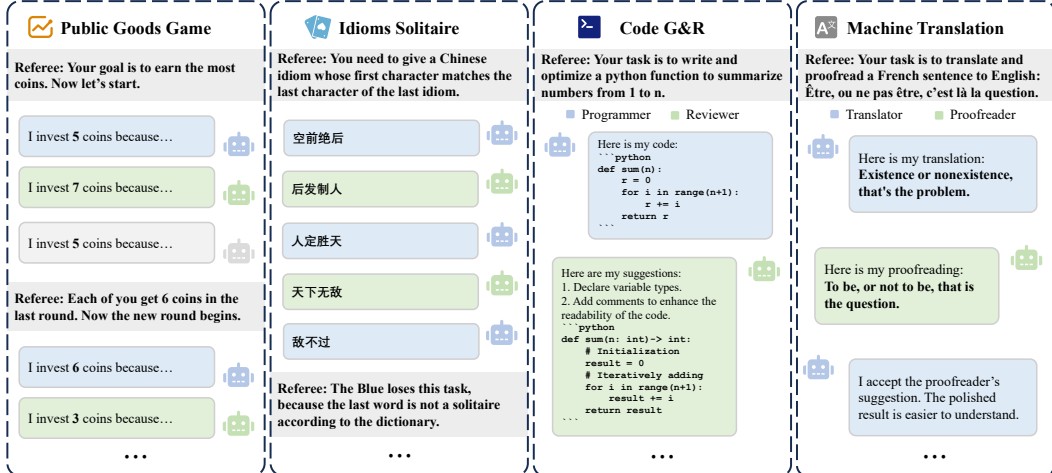

Figure 2: An overview of four implementations of dynamic interaction-based evaluation tasks.

its private goods. The PGG is a classical example in game theory, and massive studies have indicated that the PGG require the decision making ability in complex scenarios of participants to maximize their payoff. Here, we consider two task modes for the public goods game:

- **Mode 1**: After each round, the referee informs each participant the number of earnings they received in that round.

- **Mode 2**: After each round, the referee informs each participant the ordered sequence of all investment amounts for that round.

**Idiom Solitaire**. Idiom solitaire (Sun, 2012; Dobrovol'skij & Piirainen, 2010) is a symmetric evaluation task to evaluate the Chinese vocabulary of LLMs. Literally, idiom solitaire is a popular activity in China, where two LLMs give Chinese idioms alternatively, while the first Chinese character of the current idiom must be the last Chinese character of the last idiom. To win the idiom solitaire task, LLMs needs not only enough Chinese idiom vocabulary, but the ability to retrieve appropriate Chinese idioms that are not only consistent with the task rule, but difficult for other participants to retrieve the next idiom. In the idiom solitaire task, LLMs are randomly assigned the speaking order. LLMs then alternately give an idiom based on the last idiom given by other participants. The evaluation score of idiom solitaire is the number of winning of LLMs.

**Code Generation and Review**. Inspired by code generation (Yin et al., 2023; Zhang et al., 2023; Poesia et al., 2022) and code review (Li et al., 2022), Code Generation and Review (Code G&R) is an asymmetric evaluation task (i.e., participants act different roles with different action sets and goals) to evaluate the code generation ability and review ability of LLMs in real-world scenarios. Specifically, the Code G&R task requires a programmer LLM who is responsible for generating codes given natural language requirements, and a reviewer LLM who is responsible for fixing the generated codes. Then both the performances of the programmer LLM and that of the reviewer LLM are evaluated by the referee LLM. At the beginning of a Code G&R task, the referee broadcasts the description of the coding requirement to both the programmer and the reviewer. During the dynamic interaction process, the programmer and the reviewer alternatively communicates with each other through the message pool until they reach a consensus about the solution. Finally, both the performance of the programmer and the reviewer are rated by the referee.

**Machine Translation**. Machine translation (Maruf et al., 2022; Ranathunga et al., 2023) is an asymmetric evaluation task to evaluate the natural language translation ability of LLMs in real-world scenarios. In DynaEval, the machine translation consists of a translator and a proofreader. In the machine translation task, the referee first broadcast the source text and the target language. Next, the translator translates the source text to the text in the target language. Then, given the source text and the translation, the proofreader polishes the latter to facilitate its correctness and readability. Finally, both the performance of the translator and the reviewer are rated by the referee.

## 3 EXPERIMENTS

In the experiment, we selected four models for the above task, namely ChatGPT, GPT-4, Claude-2, and PaLM. For detailed information about these models, please refer to the Appendix A.6.

### 3.1 DATASETS AND EVALUATION METRICS

- **Public Goods Game**. For the two modes of this task, we conduct 10 repeated experiments for all LLMs to assess their capabilities in this task. Ultimately, we use the payoff (earning) of the LLMs at the end of the game as the evaluation metric.

- **Idiom Solitaire**. We randomly sample 30 idioms from an existing idiom database as the initial idioms and conduct experiments on all model pairs. We also swap the order of the model pairs during the experiments to evaluate the capabilities of all models under consideration. The final evaluation metric is the number of times a model wins the task.

- **Code Generation and Review**. We use a popular code generation evaluation dataset MBPP (Austin et al., 2021). For each sample in the test set, we assign each pair of models as programmer and reviewer. *To validate the effectiveness of model-based evaluation scores*, we ultimately calculated two evaluation metrics. One is the commonly used Pass@$K$ metric (Chen et al., 2021) in the code generation tasks that assess the correctness of codes through sample tests, and the other is the model-based rating score generate by the referee according to task rule. We then compare the two metrics to see whether there exists any consistency between the two metrics.

- **Machine Translation**. We select a document-level dataset (Cettolo et al., 2017) and use three language pairs for translation: English-Chinese, English-French, and German-English. We split the dataset into paragraph-level segments for the test set. For each sample in the test set, we assign each pair of models as translator and proofreader and switch roles. *To validate the effectiveness of model-based evaluation scores*, we calculate two evaluation metrics. One is the commonly used BLEU metric (Papineni et al., 2002) in machine translation tasks, and the other is the model-based rating score generate by the referee according to task rule. We then compare the two metrics to see whether there exists any consistency between the two metrics.

### 3.2 PUBLIC GOODS GAME

Evaluation results in the PGG are shown as the box plot of payoffs in Figure 3. Both mode 1 and mode 2 are run for 10 times to satisfy the stability condition. We can acquire several conclusions from Figure 3. First, in both mode 1 and mode 2, the performance of GPT-4 and Claude 2 exceeds that of PaLM and ChatGPT, which indicates that GPT-4 and Claude 2 have better decision-making ability in complex scenarios. Second, in mode 1, GPT-4 performs best, while in Mode 2, Claude 2 is the most advanced LLM. Through case study analysis, we believe this is because Claude 2 analyzes whether its previous round of investments was excessive when observing the investment situation of each participant, resulting in a relatively conservative overall investment strategy. Third, in terms of stability, GPT-4 is less stable than Claude 2 despite its better average performance. In conclusion, both the average ability and stability of LLMs differ a lot in the dynamic PGG scenario.

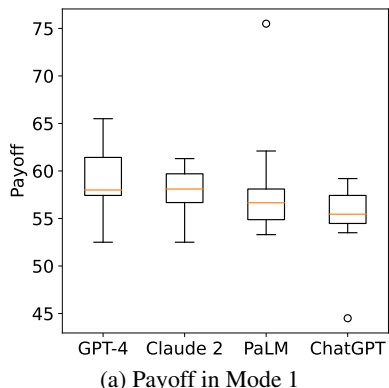

(a) Payoff in Mode 1

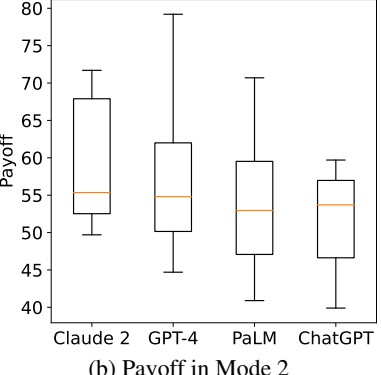

(b) Payoff in Mode 2

Figure 3: Evaluation results in PGG.

### 3.3 IDIOMS SOLITAIRE

Evaluation results in Idioms Solitaire are shown in Table 1 and Table 4. The term "Early" denotes the early position in the interaction process, while the term "Late" denotes the late position. $s_E$ and $s_L$ respectively denote the score of the early participant and the score of the late

Table 1: Evaluation results in Idioms Solitaire (winning rate).

| | | Late | | | | | | $\overline{s_E}$ |
| | | GPT-4 | | ChatGPT | | Claude 2 | | |
| | | $s_E$ | $s_L$ | $s_E$ | $s_L$ | $s_E$ | $s_L$ | |
| Early | GPT-4 | - | - | 0.33 | 0.67 | 0.57 | 0.43 | 0.45 |
| | ChatGPT | 0.75 | 0.25 | - | - | 0.78 | 0.22 | **0.77** |
| | Claude 2 | 0.3 | 0.7 | 0.25 | 0.75 | - | - | 0.28 |
| $\overline{s_L}$ | | 0.48 | | **0.71** | | 0.33 | | |

Table 2: Evaluation results in Code G&R.

| | | Prog | | | | | | | | $\overline{s_R}$ |
| | | GPT-4 | | ChatGPT | | Claude 2 | | PaLM | | |
| | | $s_P$ | $s_R$ | $s_P$ | $s_R$ | $s_P$ | $s_R$ | $s_P$ | $s_R$ | |
| Rev | GPT-4 | - | - | 8.57 | 8.83 | 8.73 | 8.77 | 8.20 | 8.72 | **8.77** |
| | ChatGPT | 8.96 | 8.60 | - | - | 8.83 | 8.89 | 8.96 | 8.73 | 8.74 |
| | Claude 2 | 8.97 | 8.73 | 8.94 | 8.78 | - | - | 8.78 | 8.72 | 8.74 |
| | PaLM | 8.99 | 8.09 | 9.04 | 8.98 | 9.01 | 8.5 | - | - | 8.52 |
| $\overline{s_P}$ | | **8.97** | | 8.85 | | 8.86 | | 8.65 | | |

participant. For example, in the first data row of Table 1, 0.33 denotes the winning rate of GPT-4 (the early position) versus ChatGPT (the late position), while 0.67 denotes that of ChatGPT. PaLM is excluded in Idiom Solitaire because it does not support Chinese input and output.

From Table 1, we can observe that the discrepancy between $\overline{s_E}$ and $\overline{s_L}$ of same LLMs are small because Idiom Solitaire is a symmetric evaluation task where different participants have the same action set and goal. Moreover, we can observe that the average winning rate and successful hit of ChatGPT are always the largest, while that of Claude 2 are always the lowest. These results demonstrate that in terms of Chinese idiom vocabulary, ChatGPT is stronger than GPT-4, and GPT-4 is stronger than Claude 2.

## 3.4 CODE GENERATION AND REVIEW

Evaluation results are shown in Table 2 and Figure 4. In Table 2, the term "Prog" denotes the programmer, and the term "Rev" denotes the reviewer. $s_p$ and $s_R$ respectively represent the scores of the programmer and the reviewer. These scores are given by the referee using a judge model (with a score range of 1-10). Different from Idioms Solitaire, Code G&R is an asymmetric task where roles of LLMs differ. As a result, the average score of an LLM as a programmer and that of the LLM as a reviewer differ more. Specifically, GPT-4 reaches the state-of-the-art performance as both of the programmer and the reviewer. ChatGPT and Claude 2 have similar coding and reviewing abilities, which are better than the ability of PaLM.

The Figure 4 further shows the performance of LLMs in different rounds of interaction. The top part shows the Pass@$K$ of LLMs as programmers *averaged on* other LLMs as reviewers, while the right part shows the Pass@$K$ of LLMs of reviewers *conditioned on* ChatGPT as the programmer. From the Figure 4 (a), we can observe that all participant LLMs *gradually improve* their code quality as the interaction goes. This observation demonstrates the significance of dynamic interaction in real-world tasks for LLMs and the potential to keep improving of LLMs. In addition, GPT-4 reaches the state-of-the-art in this evaluation task, and it also makes the highest progress in the three rounds, which illustrates its

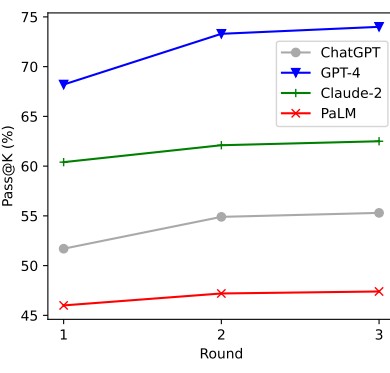

(a) Pass@$K$ of programmers.

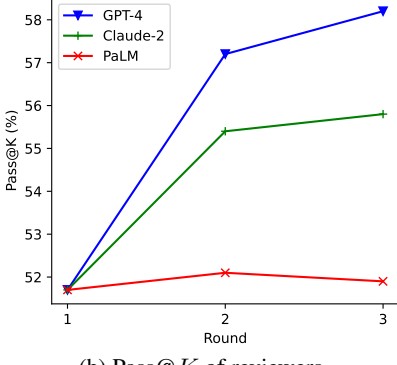

(b) Pass@$K$ of reviewers.

Figure 4: Pass@$K$ of in Code G&R.

powerful code generation ability. In Figure 4 (b), we can observe that given ChatGPT as the programmer, the improvement of code quality differs a lot as the reviewer varies. Especially, GPT-4 and Claude-2 can significantly improve the quality of codes generated by ChatGPT, while such improvement is limited if PaLM serves as the reviewer. This demonstrates the advantages of DynaEval compared to previous evaluation methods, which can not only evaluate the ability of a model for a specific static dataset but also evaluate the LLM's improvement ability based on feedback in dynamic interactive processes.

### 3.5 MACHINE TRANSLATION

Table 3: Evaluation results in Machine Translation (EN-ZH).

| | | Trans | | | | | | | | | | | | $\overline{s_{Pr}}/\overline{b_{Pr}}$ |
| | | GPT-4 | | | | ChatGPT | | | | Claude 2 | | | | |
| | | $s_T$ | $s_{Pr}$ | $b_T$ | $b_{Pr}$ | $s_T$ | $s_{Pr}$ | $b_T$ | $b_{Pr}$ | $s_T$ | $s_{Pr}$ | $b_T$ | $b_{Pr}$ | |
| | GPT-4 | - | - | - | - | 7.87 | 9.01 | 0.284 | 0.296 | 7.71 | 8.95 | 0.278 | 0.287 | 8.98/0.292 |
| Proof | ChatGPT | 7.81 | 9.08 | 0.272 | 0.296 | - | - | - | - | 7.84 | 9.09 | 0.275 | 0.293 | **9.09/0.295** |
| | Claude 2 | 7.84 | 9.05 | 0.275 | 0.290 | 7.98 | 9.0 | 0.286 | 0.293 | - | - | - | - | 9.03/0.292 |
| $\overline{s_T}/\overline{b_T}$ | | 7.83/0.274 | | | | **7.93/0.285** | | | | 7.78/0.277 | | | | |

Evaluation results in Machine Translation are presented in Table 3 (English to Chinese). Results in other languages are available at Appendix A.9. $s_T$ and $s_{Pr}$ respectively denotes scores of the translator ("Trans" in the tables) and proofreader ("Proof" in the tables) rated by the referee. These scores are given by the referee using a judge model (with a score range of 1-10). $b_T$ and $b_{Pr}$ respectively denotes the BLEU score (Papineni et al., 2002) of the translator and the proofreader. PaLM is excluded in this experiment because it supports only English. From Table 5 and Table 6, we can observe that GPT-4 reaches the state-of-the-art performance in both tasks. This result indicates that GPT-4 has a better translation and proofreading ability than ChatGPT and Claude 2. However, GPT-4 does not perform so excellent in the English to Chinese translation and proofreading. From Table 3, we can observe that ChatGPT reaches the state-of-the-art performance in the English to Chinese translation and proofreading. Indeed, this result is consistent with experiment results in Idioms Solitare, as shown in Table 1. In conclusion, considering both the aspect of idiom vocabulary and translation-proofreading, ChatGPT is the state-of-the-art LLM among the three participants, and GPT-4 ranks the second. From the experimental results, it can be seen that the ability of LLMs to play different roles is consistent. We believe this is because the abilities required for translation and polishing tasks are relatively similar.

## 4 CONCLUSION

In this paper, we studied the evaluation of large language models (LLMs) within dynamic real-world scenarios and introduced the Dynamic Interaction-based LLM-Evaluation Framework (DynaEval). We standardized the definition of the interaction process of dynamic interaction-based evaluation tasks, and we noticed that the interaction process essentially belongs to a class of dynamic games in game theory. To ensure the fairness and stability of evaluation, we introduced fairness and stability conditions for DynaEval based on properties of dynamic games. We then presented the message pool, referee, and synchronous interaction algorithm based on these studies. Furthermore, we provided four real-world scenario-based evaluation task implementations. Finally, we demonstrate the effectiveness of DynaEval through extensive experiments on the four evaluation tasks. Experiment results showed that DynaEval can effectively obtain fair and stable evaluation of the ability of various LLMs in dynamic interaction scenarios, and the dynamic interaction can indeed improve the quality of LLM outputs. For example, in the Code G&R task, the dynamic interaction between programmers and reviewers improves the correctness of generated codes, and the degree of improvement differs from reviewers to reviewers given the same programmer.

This work also has some limitations. For example, in terms of stability, DynaEval depends on multiple independent running of evaluation tasks to obtain the stable estimation of LLMs' abilities. This might be costly in some evaluation tasks with complicated interaction environments affected by too many factors. In the future, we plan to empower the efficiency of DynaEval by improving its sampling policy. Moreover, we plan to extend the range of use of DynaEval to adapt it to more real-world tasks for LLM-based applications.

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

# A APPENDIX

## A.1 RELATED WORK

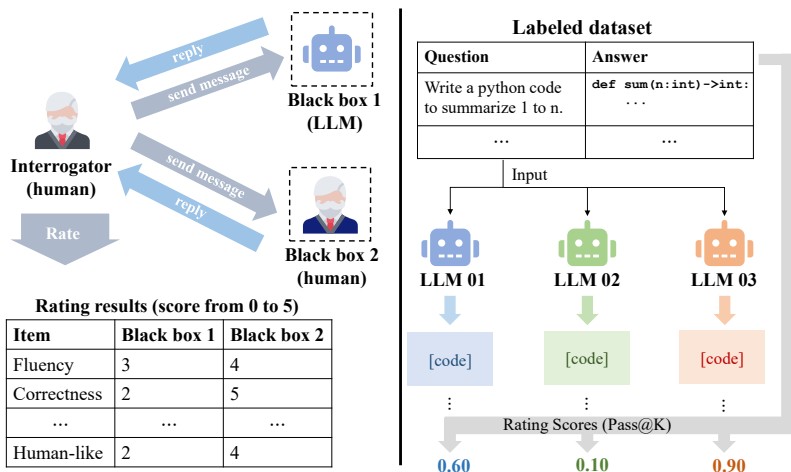

Figure 5: An illustration of existing LLM evaluation methods. **Left**: Turing test, an example of human-based evaluation methods. **Right**: Code generation, an example of supervised signal-based evaluation methods.

### A.1.1 THE EVALUATION OF LLMS

The recent advancement of LLMs many real-world tasks (e.g., code generation (Chowdhery et al., 2022), machine translation (Moslem et al., 2023)) stimulates the requirement for the evaluation of LLMs (Chang et al., 2023b). In the literature, the evaluation of LLMs can be human-based or supervised signal-based. Specifically, human-based evaluation depends on human interrogators to measure the performance of LLMs. For example, Likert scale (Petrillo et al., 2011) utilizes a rating scale filled by human judgements to measure the performance of LLMs in different dimensions. Despite their flexibility, human-based evaluations are costly and time-consuming, thus they are incapable for large-scale evaluation of LLMs. On the other hand, supervised signal-based evaluation depends on expert-labelled datasets to evaluate LLMs. Supervised signal-based evaluations are efficient, and have been applied to large-scale evaluation of LLMs in many fields such as machine translation(Bang et al., 2023b; Lyu et al., 2023; Wang et al., 2023), reasoning(Frieder et al., 2023; Saparov et al., 2023; Orrù et al., 2023) and code generation(Kashefi & Mukerji, 2023; Zhuang et al., 2023; Hendrycks et al., 2021a). Recently, there are also some studies committed to more automated and general evaluation of LLMs. For instance, MT-Bench (Zheng et al., 2023) utilizes an LLM as a judgement to automatically measure the performance of LLMs in multi-round conversations. However, conversations scenarios in this method are fixed limited to the static dataset. GameEval (Qiao et al., 2023) utilizes conversational games to gain more distinguishable evaluation results of LLMs. PromptBench (Zhu et al., 2023) focuses on the adversarial prompt resilience, and is able to evaluate the adversarial robustness of LLMs. Despite their effectiveness in some domain-specific tasks, these evaluation methods lack generality and are hard to be extended to any real-world domain.

### A.1.2 GAME THEORY

Game theory (Kreps & Wilson, 1982; Balbus et al., 2018) is a branch of mathematics that studies strategic decision-making in situations where the outcomes of individuals or groups depend on the choices made by others. It provides a framework for analysing and understanding interactions between rational decision-makers, who are assumed to pursue their own self-interest. Especially, dynamic game theory (Balbus et al., 2018) focuses on dynamic scenarios where participants can interact with others for multiple rounds and dynamically make their own decision. Key concepts in dynamic game theory include extensive-form games, which depict the sequential nature of decision-making, and concepts like Nash equilibrium and subgame perfect equilibrium. Researchers use dynamic game theory to study issues like repeated games, bargaining, and learning in strategic settings. This field has applications in various domains, including economics, biology, and computer science,

offering insights into the dynamics of strategic decision-making in complex, evolving environments. Recently, game theory has also gained attention in the research of large language models. For instance, Xu et al. (2023) utilizes an empirical study on the werewolf game to explore the performance of LLMs in communication games. Lorè & Heydari (2023) studies the strategic decision-making capabilities of GPT-3.5, GPT-4 and LLaMa-2 using four canonical two-player games. However, most existing research stays in the level of empirical study on specific game cases, which lacks generality and limits their range of application in real-world scenarios.

### A.2 PROOF OF PROPOSITION 1

In this part, we first give the formal definition of extensive games with perfect information (EGPI) (Kreps & Wilson, 1982). Next, we present the proof of Proposition 1.

**Definition 2.** *Extensive games with perfect information. An extensive games with perfect information is defined a collection of component listed as follows:*

- *A set $\mathcal{Q} = \{Q_1, Q_2, \ldots, Q_N\}$ (the set of N players).*

- *A history set $H$ of sequences (finite or infinite), which can be represented by a gaming tree. $H$ satisfies the following properties:*

  1. *The empty sequence $\emptyset$ is a member of $H$, i.e., $\emptyset \in H$, which serves as the root node of the gaming tree.*
  2. *If a sequence $(a_k)_{k=1,\ldots,K} \in H$ and $L < K$, then $(a_k)_{k=1,\ldots,L} \in H$. Further, if $(a_k)_{k=1,\ldots,K+1} \notin H$, then $(a_k)_{k=1,\ldots,K}$ is a **terminal** history.*
  3. *If an infinite sequence $(a_k)_{k=1}^{\infty}$ satisfies $(a_k)_{k=1,\ldots,L} \in H$ for every positive integer L, then $(a_k)_{k=1}^{\infty} \in H$.*

- *A terminal history set $Z$ consisting of all terminal histories, i.e., $Z = \{(a_k)_{k=1,\ldots,K} \mid (a_k)_{k=1,\ldots,K} \in H, (a_k)_{k=1,\ldots,K+1} \notin H\}$.*

- *A player function $P_f : H \backslash Z \to \mathcal{Q}$ that assigns to each non-terminal history $h \in H \backslash Z$ a member of player ($P_f(h)$ is the next payer collecting information and making decisions given the history sequence $h$).*

- *A payoff function $u_i : Z \to \mathbb{R}$ for every player $Q_i \in \mathcal{Q}$ ($u_i(z)$ is the payoff of player $Q_i$ given the terminal history $z \in Z$).*

*Then a dynamic interaction-based evaluation task can be represented as a tuple, i.e., $E = \langle \mathcal{Q}, H, Z, P_f, U \rangle$, where $U = \{u_1, \ldots, u_N\}$ is the set of payoff functions.*

*Proof.* **A Proof of Proposition 1**. Let $R^* = (G^*, C^*)$ be any interaction process of DynaEval, i.e., $R^* \in \mathcal{D}$. We aim to prove that there always exist an extensive game with perfect information $E^* = \langle \mathcal{Q}^*, H^*, Z^*, P_f^*, U^* \rangle, E^* \in \mathcal{E}$ that equivalent to $R^*$. To this end, we show that each component of $E^*$ corresponds to an equivalent component of $R^*$.

- **Player set $\mathcal{Q}^*$.** $Q^*$ indicates the number and roles of players in $E^*$. This is indeed equivalent to the LLM set $\mathcal{P}$, which is first used in the selection phase of $R^*$.

- **History set $H^*$.** $H^*$ is the set of all possible interaction histories that are terminated or non-terminated, and can be represented as a gaming tree. In this gaming tree, each node denotes an interaction output of a player given the interaction environment represented by the sequence of ancestor nodes. Indeed, any feasible path that starts from the root node ($\emptyset$) of the gaming tree represents a possible interaction history $A$ in $R^*$. Because the space of all possible interaction history is determined by interaction goal $G^*$ and interaction $C^*$, the interaction process $R^*$ indeed implicitly contains the history set $H$.

- **Terminal history set $Z^*$.** $Z^*$ is a subset of $H^*$ that defines all possible outcome of the gaming process. In the gaming tree of $H^*$, $Z^*$ represents the set of all paths that start from the root node and end at a leaf node. Similar to $H^*$, $Z^*$ is also implicitly contained by $R^*$.

- **Player function** $P_f^*$. $P_f^*(h)$ is the next player to collect information and make decisions given the history $h$. The use of $P_f^*$ in $E^*$ is equivalent to the selection phase of $R^*$ using $C^*$, where the history $h$ is equivalent to the interaction environment.

- **Payoff function set** $U^*$. $U^*$ is used to indicate the payoff (game score) of players given a terminated history. This is equivalent to scoring the performance of LLMs in the circulation phase of $R^*$, where the scoring rule is contained in the interaction rule $C^*$.

Therefore, each component of $E^*$ corresponds to an equivalent component of $R^*$. As a result, for any interaction process of DynaEval, there always exist an extensive game with perfect information that is equivalent to it. □

### A.3  THE SYNCHRONOUS INTERACTION ALGORITHM OF DYNAEVAL

The synchronous interaction algorithm of DynaEval is presented in Algorithm 1.

---

**Algorithm 1** Synchronous Interaction

---

**Input**: Interaction-based evaluation task $R$, the set of LLM participants $\mathcal{P}$
**Output**: Evaluation results $\Theta$
1: Initialize referee $P_0$
2: Initialize LLMs $P_1, P_2, \ldots, P_N \in \mathcal{P}$
3: Initialize message pool $B_{msg}$
4: **for** $i$ in $1, 2, \ldots, N$ **do**
5:     $P_0$.send($P_i$, $R$.text)                    ▷ Broadcasting the task description to LLMs
6: **end for**
7: **for** $j$ in $R$.receiveRoleSet($i_{round}$) **do**
8:     $S_{msg} \leftarrow B_{msg}$.read($P_j$, $R$, $i_{round}$)                    ▷ **Selection**: select LLMs to interact
9:     $P_0$.send($P_j$, $S_{msg}$)
10: **end for**
11: **for** $i_{round}$ in $1, 2, \ldots, M$ **do**
12:     **for** $j$ in $R$.sendRoleSet($i_{round}$) **do**
13:         $s_{msg} \leftarrow P_0$.getMessage($P_j$)        ▷ **Interaction**: LLMs interact and generate outputs
14:         $B_{msg}$.write($s_{msg}$)                        ▷ **Recording**: the interaction sequence grows
15:     **end for**
16:     **if** $P_0$.judgeEnd($R$,$B_{msg}$) is $True$ **then**
17:         Break                    ▷ **Circulation**: continue or terminate the interaction process
18:     **end if**
19: **end for**
20: Let $\Theta \leftarrow P_0$.evaluate($B_{msg}$)
21: **return** $\Theta$

---

### A.4  HOW TO DESIGN EVALUATION TASKS

The design of evaluation tasks in DynaEval is significant because it decides what to evaluate and how to evaluate. For the first aspect, the design of an evaluation task should be consistent with real-world tasks and require relevant skills such as machine translation and Code G&R. For the second aspect, the rule of the evaluation task regularizes the interaction of LLMs, thus defines how to evaluate. To this end, inspired by game thory, we propose the symmetric design and asymmetric design of evaluation tasks, which evaluate LLMs from different perspective.

- **Symmetric evaluation task**. In symmetric evaluation tasks, all LLMs act in the same role with the same action set and task goals. This can refer to symmetric games, a typical class of games in game theory, such as the prisoner's dilemma. Because the task goals of LLMs are the equal and often mutually exclusive, this type of evaluation task can evaluate the ability of LLMs in a **competitive** manner. Symmetric evaluation tasks are suitable for non-generative abilities of LLMs, such as vocabulary or decision-making.

- **Asymmetric evaluation task**. In asymmetric evaluation tasks, LLMs play different roles with different action sets and task goals. This type of evaluation task is close to real-world scenarios and can evaluate the performance of LLMs from different aspects regarding of roles they act. Especially in generative tasks such as Code G&R and machine translation, the design of asymmetric evaluation tasks can follow a *writer-editor* paradigm, which can evaluate the ability of LLMs in a **cooperative** manner. In this paradigm, there are two participants in the evaluation task totally. One LLM acts as a writer that generates outputs to meet the task requirement. The other LLM acts as an editor that fixes and polishes the writer's output to fit the task requirement better. The writing-polishing process can run for multiple rounds until the writer and the editor reach a consensus. Next, the two LLMs swap their role and repeat the task. Finally, the performance of the two LLMs can be evaluated by comparing their score on the same role, thus both the writing ability and the polishing ability can be evaluated simultaneously. Asymmetric evaluation tasks are suitable for generative abilities of LLMs, such as code generation.

## A.5 Evaluation Metrics in DynaEval

### A.5.1 Symmetric Evaluation tasks

In symmetric evaluation tasks, suppose there are $N$ LLMs. Let $V = (v_{ij})_{N \times M}$ denotes the payoff matrix calculated by the referee, where $M$ denotes the repeat times, and $v_{ij}$ denotes LLM $i$'s payoff in the $j$-th time of the task. Then all components of $V$ are comparable, because LLMs have the same role. So the evaluation result $\Theta = (\theta_1, \theta_2, \ldots, \theta_N)$ is defined as the mean score of each LLM:

$$\theta_i = \frac{1}{M} \sum_{j=1}^{M} v_{ij}, \ i = 1, 2, \ldots, N. \tag{3}$$

### A.5.2 Asymmetric Evaluation tasks

In asymmetric evaluation tasks, not all components of the payoff matrix $V$ are comparable because LLMs' roles differ. We assume that there are $L$ roles in a task ($2 \le L \le N$). Let $S = (s_{ij})_{N \times M}$ denotes the role assignment matrix, where $s_{ij} \in \{1, 2, \ldots, L\}$ denotes LLM $i$'s role in the $j$-th time of game. Then the evaluation result is a $N \times L$ matrix, i.e., $\Theta = (\boldsymbol{\theta_1}, \ldots, \boldsymbol{\theta_N}) = (\theta_{il})_{N \times L}$. The $\theta_{il}$ denotes LLM $i$'s ability when acting the role $l$. Then $\theta_{il}$ is defined as the mean score of each LLM given its role:

$$\theta_{il} = \frac{\sum_{j=1}^{M} I(s_{ij} = l) \cdot v_{ij}}{\sum_{j=1}^{M} I(s_{ij} = l)}, \ i = 1, 2, \ldots, N, \tag{4}$$

where $I(\cdot)$ denotes the indicator function.

## A.6 LLMs to Evaluate

- **ChatGPT**. ChatGPT is a large language model developed by OpenAI that can effectively follow various human instructions. The model version used in our experiments is "gpt-3.5-turbo-0613."

- **GPT-4** (OpenAI, 2023). GPT-4 is OpenAI's most advanced system, with stronger conversational and reasoning capabilities compared to ChatGPT. It supports longer context inputs and performs at a human-level across multiple tasks, with higher accuracy and greater creativity and collaboration. The model version used in our experiments is "gpt-4-0613."

- **Claude 2** (Wu et al., 2023). Claude 2 is a large language model developed by Anthropic. It has enhanced capabilities in code writing, text analysis, and mathematical reasoning. Claude 2 uses a technique called "natural induction" for training, which allows the model to infer general rules from a few examples. The model version used in our experiments is "claude-2."

- **PaLM** (Chowdhery et al., 2022). PaLM is a large language model developed by Google, with 540 billion parameters. It excels in code writing, text analysis, and mathematical reasoning, and has strong generative capabilities. The model version used in our experiments is "chat-bison-001."

## A.7 INPUT RULE OF DIFFERENT LLMs

Both ChatGPT and GPT-4 support role-based messages as input. Users can create messages for three different roles, including "system," "user," and "role." Therefore, for models like ChatGPT and GPT-4, their own utterances should be stored as "assistant messages" in the history, while all other participants' utterances, including those of the host, should be stored as "user messages." For the LLMs Claude 2 and PaLM, messages are distinguished using human tags and role tags. The human tag and role tag for Claude 2 are "\n\nHuman:" and "\n\nAssistant:", respectively. For PaLM, the human tag and role tag are "\n\nUser:" and "\n\nBot:", respectively.

## A.8 PROMPTS OF EVALUATION TASKS

- Public Goods Game (PGG): For each participant, the initial prompts used are as follows:

  Mode 1: "You are playing a game, there are n participants in this game, a total of 5 rounds. At the beginning of each round, you will get 10 coins. You can choose to invest X ($0 \leq X \leq 10$) coins in the public pool. At the end of each round, the coins in the public pool will be multiplied by $\alpha$ times and be evenly distributed to all participants. Your goal is to have the most coins at the end of the game."

  Mode 2: "You are playing a game, there are n participants in this game, a total of 5 rounds. At the beginning of each round, you will be informed of the amount of invested coins in the last round, which is represented by a descending sorted list where each element is the amount of a player's invested coins. Then you will get 10 coins, you can choose to invest X ($0 \leq X \leq 10$) coins in the public pool. At the end of each round, the coins in the public pool will be multiplied by $\{\alpha\}$ times and be evenly distributed to all participants. Your goal is to have the most coins at the end of the game."

  After the end of round $i$, each participant receives the following prompt:

  Mode 1: "In the last round you earned {income} coins. The $i + 1$ round starts, you get 10 coins. Please give the amount of coins you want to invest and explain your decision reason." Here, 'income' is calculated as the total investment in round $i$ multiplied by $\alpha/n$, and 'sorted invest' represents the ascending set of all players' investment amounts in round $i$.

  Mode 2: "In the last round the amount of invested coins is sorted invest where each element is the amount of a player's invested coins. The $i + 1$ round starts, you get 10 coins. Please give the amount of coins you want to invest and explain your decision reason."

  After receiving each participant's response, the referee uses the following prompt to format the reply: "Your output must follow the json format below: {"reason":"⟨reason⟩", "coins":⟨Investment amount⟩}"

- Idiom Solitaire: In round i, the prompt received by the participant is:

  "You are participating in an idiom chain game. In this game, you need to give a four-character idiom where the first character matches the last character of the previous idiom. The idioms used in the same game cannot be repeated. If the first character of your output is incorrect or if it is not a Chinese idiom, your opponent wins. Your ultimate goal is to win the game. Next, I will provide you with the context of the current idiom chain and connect them using '→'. Please provide an appropriate idiom that follows these rules. Please note that you only need to provide the idiom without any other response. $\{S_I\}$" Here, $S_I$ represents the sequence of idioms used in the idiom solitaire task so far, connected by '→'.

- Code Generation and Review (Code G&R): For the Programmer, the initial prompt is constructed based on the programming question:

  "You will play the role of a programmer, and you need to solve various programming problems provided by users, and provide complete and executable solutions. Remember, you only need to give pure Python code without any extra explanation. Question: $\{Q\}$" Here, $Q$ represents the current programming problem.

  In subsequent tasks, the prompt is constructed based on the Reviewer's code comments:

  "Reviewer: $\{C_R\}$ Please give a revised solution based on the following review comments. Remember, you only need to give pure Python code without any extra explanation." Here $C_R$ represents the Comments of the Reviewer.

  For the Reviewer, the initial prompt is constructed based on the Programmer's response to the programming task:

"You will play the role of a code reviewer. You need to review the code provided by the programmer and give your feedback. You must comment on the nature of the code in three aspects: Code correctness, Code clarity, and Efficiency. Question: $\{Q\}$ Programmer: $\{A\}$" Here, $A$ represents the current round Programmer's response.

The subsequent prompt used is:

"Please continue to submit review comments according to the improved procedure. If you think the programmer performs well in three aspects, please just output 'over' without any other output. Programmer: $\{A\}$"

In the task of code generation, a judge model is employed to evaluate the performance of both the Programmer and the Reviewer based on a specific prompt:

"You will play the role of a professional developer, and you will rate both a programmer and a reviewer based on their conversation. The main criteria are: 1. Whether the code provided by the programmer meets the requirements of the problem. 2. Whether the programmer has improved the code according to the reviewer's suggestions. 3. Whether the reviewer has given reasonable and actionable feedback for improvement. Please note: 1. Points are given on a scale of 1-10. 2. You need to give not only a final grade, but also a specific basis for the grade. 3. Please reply using the following format: {"Programmer":{"evaluation":⟨explain⟩, "score":⟨ score ⟩}, "Reviewer":{"evaluation":⟨explain⟩, "score":⟨score⟩}}"

- Machine Translation: For the Translator, the prompt is constructed based on the current source language, target language, and content to be translated:

"You will play the role of a professional translator. Please translate the given text from $\{L_s\}$ (source language) to $\{L_t\}$ (target language). Source language text: $\{T_s\}$" Here, $L_s$ represents the source language, $L_t$ represents the target language, and $T_s$ represents the content that needs to be translated.

For the Proofreader, the prompt is constructed based on the Translator's output:

"You will play the role of a professional translation editor. Your task is to polish the $\{L_t\}$ translation provided by the translator for the given text in $\{L_s\}$ (source language), making the translated content more accurate. Note that you only need to reply with the polished sentence in the target language, not any other reply. Source language text: $\{T_s\}$ Translator: $\{T_t\}$" Here, $T_t$ represents the content translated by the Translator.

In machine translation, the referee uses a judge model to evaluate translator and proofreader performance. The prompt used by the judge model is:

"You will play the role of a translation expert, and you will rate the dialogue between the translator and the proofreader. The main criteria are: 1. Whether the translator's translation of the given text is semantically consistent with the original text. 2. Whether the proofreader's polishing result of the translator's translation is more accurate. Please note: 1. Points are given on a scale of 1-10 2. Not only do you need to give a final grade, but you also need to give a specific basis for the grade 3. Please reply using the following json format: {"Translator":{"evaluation":⟨explain⟩, "score":⟨score⟩}, "Proofreader":{"evaluation":⟨explain⟩, "score":⟨score⟩}}"

## A.9 MORE DETAILED EVALUATION RESULTS

This part shows more detailed evaluation results in idioms solitaire and machine translation.

Table 4: Evaluation results in Idioms Solitaire (successful hit).

| | | Late | | | | | | $\overline{s_E}$ |
| | | GPT-4 | | ChatGPT | | Claude 2 | | |
| | | $s_E$ | $s_L$ | $s_E$ | $s_L$ | $s_E$ | $s_L$ | |
| Early | GPT-4 | - | - | 0.89 | 1.11 | 1.29 | 1.14 | 1.09 |
| | ChatGPT | 1.12 | 0.75 | - | - | 1.11 | 0.78 | **1.12** |
| | Claude 2 | 0.8 | 1 | 0.75 | 1.25 | - | - | 0.78 |
| | $\overline{s_L}$ | 0.88 | | **1.18** | | 0.96 | | |

Table 5: Evaluation results in Machine Translation (DE-EN).

| | | Trans | | | | | | | | | | | | $\overline{s_{Pr}}/\overline{b_{Pr}}$ |
|---|---|---|---|---|---|---|---|---|---|---|---|---|---|---|
| | | GPT-4 | | | | ChatGPT | | | | Claude 2 | | | | |
| | | $s_T$ | $s_{Pr}$ | $b_T$ | $b_{Pr}$ | $s_T$ | $s_{Pr}$ | $b_T$ | $b_{Pr}$ | $s_T$ | $s_{Pr}$ | $b_T$ | $b_{Pr}$ | |
| | GPT-4 | - | - | - | - | 8.24 | 9.26 | 0.405 | 0.417 | 8.09 | 9.28 | 0.405 | 0.419 | **9.27/0.418** |
| Proof | ChatGPT | 8.27 | 9.26 | 0.410 | 0.421 | - | - | - | - | 8.18 | 9.23 | 0.401 | 0.409 | 9.25/0.415 |
| | Claude 2 | 8.26 | 9.24 | 0.407 | 0.412 | 8.19 | 9.18 | 0.406 | 0.414 | - | - | - | - | 9.21/0.411 |
| $\overline{s_T}/\overline{b_T}$ | | **8.27/0.409** | | | | 8.22/0.406 | | | | 8.14/0.403 | | | | |

Table 6: Evaluation results in Machine Translation (EN-FR).

| | | Trans | | | | | | | | | | | | $\overline{s_{Pr}}/\overline{b_{Pr}}$ |
|---|---|---|---|---|---|---|---|---|---|---|---|---|---|---|
| | | GPT-4 | | | | ChatGPT | | | | Claude 2 | | | | |
| | | $s_T$ | $s_{Pr}$ | $b_T$ | $b_{Pr}$ | $s_T$ | $s_{Pr}$ | $b_T$ | $b_{Pr}$ | $s_T$ | $s_{Pr}$ | $b_T$ | $b_{Pr}$ | |
| | GPT-4 | - | - | - | - | 7.90 | 9.15 | 0.326 | 0.343 | 7.84 | 9.12 | 0.324 | 0.338 | **9.27/0.341** |
| Proof | ChatGPT | 8.02 | 9.01 | 0.338 | 0.351 | - | - | - | - | 7.89 | 9.15 | 0.327 | 0.345 | 9.25/0.348 |
| | Claude 2 | 8.07 | 9.01 | 0.344 | 0.357 | 8.09 | 8.98 | 0.321 | 0.337 | - | - | - | - | 9.21/0.347 |
| $\overline{s_T}/\overline{b_T}$ | | **8.27/0.341** | | | | 8.22/0.324 | | | | 8.14/0.326 | | | | |

## A.10 CASE STUDY

### A.10.1 A CASE WITH ILLUSTRATION

In Figure 6, we separately show case examples of Idiom Solitaire and Machine Translation to help us better understand how our framework evaluates the capabilities of models in these tasks. Detailed cases are shown in Appendix A.10.2. For Idiom Solitaire, the model primarily needs to know what an idiom is and understand the rules of the Idiom Solitaire task. In the example, Claude 2 fails to come up with an idiom that starts with the Chinese character resulting in a failed chain.

For Machine Translation: As a translator, the model needs to translate a paragraph from the source language into the target language, while the proofreader needs to improve the translation. In the example, GPT-4, acting as the translator, accurately captured the meaning of the original text but still had some minor issues. Claude 2, serving as the proofreader, effectively improved GPT-4's translation. Using the word "perform" instead of "operate" in the sentence more accurately restored the original text's semantics.

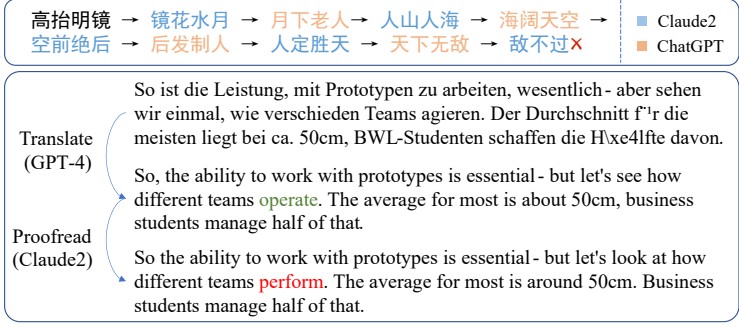

Figure 6: Cases of Idioms Solitaire and Machine Translation.

### A.10.2 OTHER CASES

In Figure 7 and 8, we observed that different models adopted different investment strategies in the PGG task. GPT4 takes a more conservative approach by gradually increasing investments instead of blindly adding to them. Conversely, PaLM and ChatGPT adopt more aggressive strategies, investing all coins into the item pool in the second round after earning profits from their initial investment plan.

In Figure 9, we identified three reasons for task termination in the idiom solitaire task: 1. The final output content length is not a four-character idiom; 2. The output is a four-character word rather than an idiom; 3. The output is a correct idiom but does not match with the first and last characters of the previous idiom.

In Figure 10, we can observe that in machine translation tasks, when the translation by the translator is not satisfactory, the proofreader will improve and enhance the final translation result. However,

there are instances where the Translator performs well initially but quality decreases after modifications by the proofreader.

In Figure 11, we can observe that in Code G&R tasks, when the programmer demonstrates insufficient understanding of Perrin definition; the Reviewer identifies errors and provides correct definitions while evaluating code correctness; on the other side, the programmer may write correct but suboptimal code; in such cases reviewer may offer time complexity analysis and suggest better implementation approaches. This task assesses the programmer's programming abilities as well as the Reviewer's judgment and suggestions regarding code correctness, readability, and time complexity.

**step: 0**

GPT-4: I will invest 5 coins. Reason: As this is the first round, it's hard to predict the investment behavior of other participants. So, I choose to invest half of my coins, which is a balanced strategy. This way, I can still have some coins left if the return from the public pool is not good, and also have the chance to get more coins if the return is good.

PaLM: I would like to invest 5 coins in the first round. This is because I believe that the game is fair and that I have a good chance of winning. I also believe that the other players are likely to invest a similar amount of coins, so I am not at a disadvantage by investing less.
Overall, I believe that investing 5 coins in the first round is the best decision for me.

Claude2: Okay, for the first round I have 10 coins to invest. I don't have any prior information to guide my investment decision, so I will start cautiously by investing 5 coins. This allows me to invest a reasonable amount while still retaining some coins in case I lose on the investment. As the rounds continue and I get more information, I can adjust my investment amount accordingly. For now, I will invest 5 coins and explain that I am starting small as I have no data yet on potential returns or risks.

ChatGPT: I will invest 5 coins in the public pool. My decision is based on the fact that I want to have a balance between investing in the public pool and keeping some coins for myself. By investing 5 coins, I am contributing to the growth of the public pool, which will benefit me in the long run when the coins are distributed evenly among all participants. Additionally, by keeping 5 coins for myself, I ensure that I have some coins to use in future rounds.

**step: 1**

GPT-4: I will invest 6 coins. Reason: Since all players invested equally in the last round, it seems like everyone is playing conservatively. By investing slightly more this round, I can potentially gain a larger share of the public pool. However, I'm still keeping a portion of my coins to ensure I have some left in case the return is not as high as expected.

PaLM: Based on the information you have provided, I would like to invest 10 coins in the second round. This is because I believe that the other players are likely to invest a similar amount of coins, so I am not at a disadvantage by investing less.

Claude2: In the first round of the game, every player, including myself, invested 5 coins, leaving me with little additional insight. For the second round, I plan to invest 7 coins, slightly increasing my stake based on my initial experience. I'll keep 3 coins uninvested as a safeguard. This cautious approach allows me to seek higher returns while retaining flexibility for future adjustments as I gather more data on the game's risks and returns.

ChatGPT: In the previous round, all players invested equally, leading to an even distribution from the public pool. To optimize my winnings, I plan to invest all 10 of my coins into the public pool for the next round. This will not only increase the total pool but also boost my share when it's distributed. This all-in strategy aims to capitalize on the pool's growth for potentially higher returns in the long run.

**step: 2**

GPT-4: I will invest 7 coins. Reason: The last round showed that other players are willing to invest more, which means the return from the public pool could be higher. I will increase my investment slightly to try to gain a larger share. However, I will still keep some coins in reserve in case the return is not as expected. This strategy allows me to potentially gain more while still maintaining some level of safety.

PaLM: Based on the information you have provided, I would like to invest 8 coins in the third round. This is because I believe that the other players are likely to invest a similar amount of coins, so I am not at a disadvantage by investing less.
I would also like to point out that the amount of coins I invest is not the only factor that determines my chances of winning. The other players' strategies, the luck of the draw, and the overall structure of the game also play a role. Therefore, it is important to be flexible and to adapt my strategy as the game progresses.

Claude2: In the previous round, players including the author invested [10, 10, 7, 6] coins, with the author investing 7 coins. Some players went all-in with 10 coins, indicating potential upside. However, since not all players invested fully, uncertainty remains.
For the 3rd round, the author will invest 8 coins out of 10 received. This represents a gradual increase in investment confidence.
The author aims to strike a balance between capitalizing on perceived investment potential while exercising caution due to potential unknown risks. The author will hold back from investing the full 10 coins until observing consistent heavy investments from other players.

ChatGPT: In the previous round, players invested different amounts of coins. For this upcoming round, I'll consider two main factors. Firstly, I want to outperform others by having the most coins, as the ultimate winner is the one with the highest coin count. Secondly, I aim to strike a balance between investing in the communal pool and holding onto coins for myself. Contributing to the communal pool can lead to future benefits, but I must also reserve enough coins for future rounds.
My decision is to invest 6 coins in the communal pool. This choice is influenced by the fact that the player who invested the least in the previous round had 6 coins. I'm aiming to maintain competitiveness by matching their investment while also retaining coins for myself.

Figure 7: Cases of PGG (Part 1).

**step: 3**

GPT-4: I will invest 8 coins. Reason: The last round showed that other players are still investing a significant amount, which means the return from the public pool could be high. By investing slightly more this round, I can potentially gain a larger share of the public pool. However, I'm still keeping a portion of my coins to ensure I have some left in case the return is not as high as expected. This strategy allows me to potentially gain more while still maintaining some level of safety.

PaLM: Based on the information you have provided, I would like to invest 9 coins in the fourth round. This is because I believe that the other players are likely to invest a similar amount of coins, so I am not at a disadvantage by investing less. I would also like to point out that the amount of coins I invest is not the only factor that determines my chances of winning. The other players' strategies, the luck of the draw, and the overall structure of the game also play a role. Therefore, it is important to be flexible and to adapt my strategy as the game progresses.

Claude2: In the prior round, investments from all players were fairly cautious, ranging between 6 and 8 coins, with my own being 8 coins. Given that no one has fully committed with a 10-coin investment for consecutive rounds, uncertainty about downside risk remains. Consequently, I will cautiously increase my investment to 9 coins for the 4th round. This strategy aims to give me a slight edge in earnings potential while minimizing risk. My approach is conservative and incremental, allowing room for adjustments as I gather more data on player behavior and market dynamics. In summary, I'll invest 9 coins this round to balance risk and reward, reserving the option to change tactics as more information becomes available.

ChatGPT: In the previous round, the lowest investment was 6 coins, while most players invested 7 or 8 coins. Taking into account this distribution and the potential for growing the public pool, I've decided to invest 8 coins for the current round. My aim is to align my investment with the majority to maintain competitiveness and to capitalize on the potential for larger distributions in future rounds. This strategy seeks to optimize my returns while keeping me in a competitive position.

**step: 4**

GPT-4: I will invest 10 coins. Reason: This is the last round and the previous rounds have shown that the other players are consistently investing a high number of coins. By investing all my coins, I have the chance to maximize my return from the public pool. Since there are no more rounds after this, there's no need to keep any coins in reserve.

PaLM: For the fifth round, I've decided to invest all 10 of my coins, as the pattern of previous rounds suggests other players are likely to do the same. This decision aims to keep me competitive and capitalize on any upside potential. However, I acknowledge that additional information on other players' strategies could influence my decision in future rounds. For example, if another player is known for aggressive investments, I might opt for a more conservative approach. It's worth noting that the game's outcome is not solely determined by the size of my investment. Factors like the strategies of other players, luck, and game mechanics also influence the result. Therefore, I aim to remain flexible and adjust my strategy as the game evolves.

Claude2: In the last round, investments were between 8 and 9 coins, and I invested 9. The consistent but cautious increases from all players indicate perceived upside potential, yet no one has gone all in. For the 5th round, I've decided to invest all 10 of my coins. The pattern of gradual investment increases gives me confidence that there is untapped upside potential. While there's still some risk involved, I believe the potential returns now outweigh it. In summary, the cautious yet rising investments from previous rounds have convinced me to fully invest, aiming to capitalize on what appears to be a favorable risk-reward scenario. If others also go all in, it will further validate my assessment.

ChatGPT: In the previous round, player investments ranged between 8 and 9 coins. Given this and the opportunity to grow the public pool, I've decided to invest all 10 coins for this final round. My decision is influenced by the fact that the majority of players previously invested 9 coins. By investing the maximum amount, I aim to maximize my potential returns and improve my chances of ending the game with the most coins. This strategy seeks to capitalize on both the current investment patterns and the final opportunity to influence the public pool's distribution.

Figure 8: Cases of PGG (Part 2).

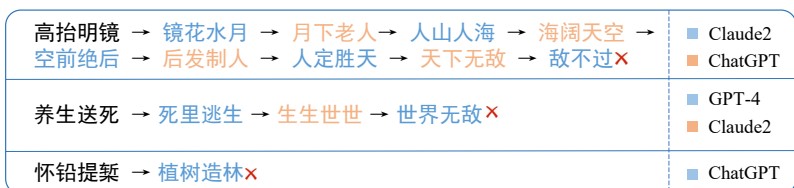

Figure 9: Cases of Idioms Solitaire.

Source Text：We're evolving to be a more collaborative and hearty species. This is true. I believe this. So, consider this really interesting statistic; it was recently published by a researcher at Carnegie Mellon University: The average young person today in a country with a strong gamer culture will have spent 10,000 hours playing online games by the age of 21. Now 10,000 hours is a really interesting number for two reasons. First of all, for children in the United States, from fifth grade to high school graduation, if you have perfect attendance.

GPT-4：我们正在演化为一个更具合作精神和活力的物种。这是真的，我相信这一点。那么，考虑这个非常有趣的统计数据；最近由卡内基梅隆大学的一位研究者发布：在一个拥有强大游戏文化的国家，今天的平均年轻人到21岁时将会花费10,000个小时在线玩游戏。现在，10,000个小时是一个非常有趣的数字，有两个原因。首先，对于美国的孩子，从五年级到高中毕业，如果你的出勤率完美无缺。

ChatGPT：我们正在演化成为一个更加合作和有活力的物种。这是确实的，我相信这一点。所以，考虑一下这个非常有趣的统计数据吧；最近，卡内基梅隆大学的一位研究员发布了一项数据：在一个有着强烈游戏文化的国家，如今的年轻人在21岁之前平均会花费10,000个小时去玩在线游戏。现在，10,000个小时这个数字非常有趣，有两个原因。首先，对于美国的孩子来说，从五年级到高中毕业，在校期间如果你没有缺席的话，

gold：我们正在不断发展成为一个更具有协作精神和更强健的物种，这是真的，我深信不疑。来看一个有趣的调查，这份调查结果是由一位来自卡内基梅隆大学的研究者发布的，在一个玩游戏氛围很强的国家，当今的年轻人 在一个玩游戏氛围很强的国家，当今的年轻人 在其21岁前，平均每人花费在网络游戏上的时间可达10000小时 现在，一万小时是一个很有趣的数值，主要是因为有两个原因：首先，美国的儿童从五年级到高中毕业所需要的时间就是10080个小时，当然你得表现良好（按时上课，出勤）

Source Text：That's not something we have in real life that easily, this sense that at our fingertips are tons of collaborators. And there's this epic story, this inspiring story of why we're there, and what we're doing, and we get all this positive feedback. You guys have heard of leveling up, +1 strength, +1 intelligence. We don't get that kind of constant feedback in real life. When I get off this stage, I'm not going to have +1 speaking, and +1 crazy idea, +20 crazy idea.

ChatGPT：这不是我们在现实生活中轻易拥有的东西，这种感觉在我们的指尖上有大量的合作者。还有这个史诗般的故事，这个鼓舞人心的故事告诉我们为什么我们在这里以及我们正在做什么，我们得到了所有这些积极的反馈。你们可能听说过升级，+1 力量，+1 智力。我们在现实生活中没有那种持续的反馈。当我下台后，我的演讲技巧不会变成+1，我的疯狂想法也不会增加+1 或是+20。

GPT-4：这不是我们在现实生活中容易获得的，那种只要动动手指，就能找到许多合作伙伴的感觉。还有那个宏大的故事，那个触动人心的故事，向我们诉说我们为何身处此地，我们在做什么，我们收到所有这些积极的反馈。你们可能已经听说过升级，+1力量，+1智力这样的事情。然而，我们在现实生活中是不会得到那种持续的反馈的。当我从这个讲台上下来时，我并不会因为此次演讲而+1演讲技巧，也不会因为疯狂的想法而+1或者+20。

gold：在现实生活中就没有这么轻松了，这种在我们指尖间就有成千上万合作伙伴的感觉，在现实生活中就没有这么简单了。在游戏中会有美妙的故事情节，而这些情节 吸引了我们，并且激励着我们的所作所为。同时，对于我们所做出的任何努力都会有一个积极肯定的回应。你们肯定听说过等级提升， 伴随着等级提升力量加一点同时智力加一点 而我们在现实生活中就没有这种持续的回馈。比如说，当我走下这个舞台的时候，我不会有演说能力加一，疯狂的创意能力加一，更不用说疯狂的创意能力加20了。

Figure 10: Cases of Machine Translation.

**Problem：Write a function to calculate the sum of perrin numbers.**

PaLM:
```python
def perrin(n):
    if n == 0:
        return 0
    elif n == 1:
        return 2
    else:
        return perrin(n - 2) + perrin(n - 1) - 2

def sum_perrin(n):
    sum = 0
    for i in range(n + 1):
        sum += perrin(i)
    return sum
```

GPT-4：
Code Correctness: The implementation of the perrin sequence in the code is incorrect. The recursive formula does not subtract 2, instead it is p(n- 2) + p(n - 3) for n>=3 wherein p(0)=3, p(1)=0, p(2)=2. Hence, the existing implementation of 'perrin' function cannot produce correct results.
Code Clarity: The code is pretty straightforward, readable and easy to understand. Function and variable names are explanatory. However, the incorrect implementation of the perrin sequence detracts from its overall clarity.
Efficiency: The current implementation computed the perrin numbers in a recursive way which can cause an exponential amount of function calls for larger inputs. This is inefficient and can lead to a very slow runtime and even a stack overflow for large inputs. Caching previously computed values or using an iterative approach to compute the perrin numbers would greatly improve efficiency.

PaLM:
```python
def perrin(n):
    if n == 0:
        return 3
    elif n == 1:
        return 0
        elif n == 2:
                return 2
    else:
        return perrin(n - 2) + perrin(n - 3)

def sum_perrin(n):
    sum = 0
    for i in range(n + 1):
        sum += perrin(i)
    return sum
```

GPT-4: over

**Problem：Write a python function to find the number of divisors of a given integer.**

Claude2:
```python
def count_divisors(n):
  count = 0
  for i in range(1, n+1):
    if n % i == 0:
      count += 1
  return count
```

ChatGPT:
Code correctness: The code is correct in terms of finding the number of divisors of a given integer. It uses a for loop to iterate through all numbers from 1 to n, and checks if n is divisible by each number. If it is, the count is incremented.
Code clarity: The code is clear and easy to read. The variable names are descriptive, and the purpose of each line is clear.
Efficiency: The code has a time complexity of O(n) because it iterates through all numbers from 1 to n. This can be improved to O(sqrt(n)) by iterating only up to the square root of n, as factors above the square root will already be accounted for by factors below the square root.
To improve efficiency, you can modify the for loop to iterate up to the square root of n, and count both divisors below and above the square root separately. You can then return the count multiplied by 2 to account for both sets of divisors.

Claude2:
```python
import math

def count_divisors(n):
  count = 0
  for i in range(1, int(math.sqrt(n)) + 1):
    if n % i == 0:
      count += 1
  count *= 2
  return count
```

ChatGPT : over

■ Programmer    ■ Reviewer

Figure 11: Cases of Code G&R.

