# OpenReview forum: "DynaEval: A Dynamic Interaction-based Evaluation Framework for Assessing LLMs in Real-world Scenarios"
_ICLR.cc/2024/Conference — Submitted to ICLR 2024_

### Official Review · Reviewer_EKE2 · 2023-11-01

**Soundness:** 2 fair
**Presentation:** 2 fair
**Contribution:** 1 poor
**Rating:** 3
**Confidence:** 2

**Summary:**

The authors propose a novel framework for evaluating LLMs in goal-driven, multi-agent, and multi-turn settings called DynaEval. They motivate their approach by pointing out the gap between the existing benchmarks and how LLMs will be used in the wild, which is in interaction with other humans/agents/LMs in which they try to achieve a final goal (be it a common goal or a personal one).

In the first step, they propose what they believe to be the common ground of such real-world interaction processes to create a unified definition. They call this the Interaction process of DynaEval, which mainly consists of an interaction goal and an interaction rule. They then draw a relationship between this interaction process and extensive games with perfect information (not described in the paper). Furthermore, they then enforce the anonymity and perfect information rules from EGPI on the DynaEval interaction process in order to overcome the stability and fairness issues that they identified.

Next, they explain the structure of the DynaEval process implementation, which consists of a referee, a message pool, and a synchronous interaction algorithm. The authors then explain the implementation of 4 games to evaluate the LLMs with.

Finally, they do experiments with to compare 4 existing LLMs in the aforementioned games under the DynaEval process. The authors claim that GPT-4 is superior in 3 out of the 4 games and ChatGPT (I'm assuming that means GPT-3.5?) is superior in the remaining one.

**Strengths:**

- Attempting to go beyond the simple static framework of evaluation for LLMs, trying to bring the evaluation setting closer to reality of how these models will be used
- I quite liked the idea of using games to evaluate LLMs, since even though the games can be quite complex, the evaluation criteria can be pretty simple and mechanistic

**Weaknesses:**

1. My first issue with the paper is clarity:

    a. Even though EGPI seems to be a central concept in their paper, the authors do not even explain what it is. They simply brush over it by adding a citation. I wouldn't expect more than a small minority of the audience at ICLR to be familiar with this concept.

    b. The DynaEval interaction process doesn't seem quite natural and it's not explained from where the "interaction goal" and "interaction rule", which are supposed to be defining characteristics of these processes, come from. If I had to guess, I would think that the "definition" came as a result of trying to match EGPI definition with the DynaEval process and not the other way around.\

    c. The equations (1) and (2) seem totally out of place in the paper. Honestly, I'm not sure what role they play in explaining any of the concepts. Are they just there to explain the concept of sampling, or are they there simply to provide a "scare factor"?

2. Another point of contention is statistical significance:

    a. Looking at Figure 3. Both Figures 3.a and 3.b show a set of highly overlapping distributions. The medians of the top contenders in 3.a are barely distinguishable even, however the conclusion that section 3.2 notes is that GPT-4 performs best is mode 1. It seems to me like the authors are making claims without regard to statistical significance of their results.
    b. The same is true in Table 2: the difference between GPT-4 (8.77) and ChatGPT (8.74) seems negligible, but that is not reflected in the conclusions of the text


3. I'm not sure of the utility of many parts of the proposal:

    a. I'm not convinced that DynaEval framework itself brings anything to the table. I get the utility of using games as evaluation methods, which is a good idea depending on the implementation, but the framework itself doesn't quite bring anything to the table IMO.

    b. The games seem odd, and it's not obvious what capability of the model they are trying to measure. For example, I'm not even sure it makes sense for the Code G&R and the Machine Translation games to actually be turn-based games. As I said, I do see why games can be useful, but the devil is in the details

**Questions:**

I ended up aggregating questions in the weaknesses section as it made more sense.

---

> ### Author Response · Authors · 2023-11-13
> **Reply to Reviewer EKE2 (1-Part 1)**
>
> We would like to thank you very much for the detailed feedback and valuable suggestions.
>
> The “ChatGPT” in experiments means GPT-3.5-turbo. Sorry for the confusion. We will clarify this in the new version of our paper.
>
> W1.a. Even though EGPI seems to be a central concept in their paper, the authors do not even explain what it is. They simply brush over it by adding a citation.
>
> A1. a. Thank you for your question. We explained EGPI in our paper in p3 (after the citation), and introduced the formal definition of EGPI in Appendix A.2. Due to the page limitation, we did not place the definition and detailed explanation of EGPI in the main body of this paper.
>
> W1.b. The DynaEval interaction process doesn't seem quite natural and it's not explained from where the "interaction goal" and "interaction rule", which are supposed to be defining characteristics of these processes, come from.
>
> A1.b. As mentioned in the first paragraph of Section 2.2, it should be noticed that the definition of dynamic interaction is totally based on our observation of real-world dynamic interaction rather than matching with EGPI. The relationship between DynaEval and EGPI is an accidental discovery, and we find that EGPI can help us to standardize the evaluation of LLMs. Indeed, the “interaction goal” and “interaction rule” are common denominators of dynamic interactions and are sufficient to depict the former. For example, the review-rebuttal process of ICLR is a dynamic interaction scenario, which can be completely depicted by interaction goals (of reviewers and authors) and interaction rules. For reviewers, their interaction goal is to accept high-quality papers and reject low-quality papers and to help improve the quality of papers. For authors, their interaction goal is to explain to and discuss with reviewers to maximize the probability of publishing their paper at this conference and also improve the quality of their paper. The interaction rule of the review-rebuttal process is that reviewers first give their original reviews of submitted papers, and then reviewers and authors can freely discuss the paper during the rebuttal phase.
>
> W1.c. The equations (1) and (2) seem totally out of place in the paper. Honestly, I'm not sure what role they play in explaining any of the concepts. Are they just there to explain the concept of sampling, or are they there simply to provide a "scare factor"?
>
> A1.c. Thank you for your doubt. The equations (1) and (2) aim to model and alleviate the uncertainty of LLMs’ performance in dynamic interaction-based evaluation, which is significant in LLM evaluation. As shown in our experiments, the performance of LLMs is unstable and can fluctuate given different interaction environments. Unlike traditional supervised signal-based evaluations where the environment (context) is static, the environment of dynamic interaction-based evaluation is itself dynamic, thus magnifying the uncertainty of LLMs’ performance. Therefore, it is indispensable to formally model the performance of LLMs with “uncertainty” and to alleviate the effect of uncertainty on evaluation results. As a result, rather than modeling the performance of LLMs by deterministic values, we find that modeling by random variables is more suitable for this problem and utilize equations (1) and (2) to present how our method models and alleviates the uncertainty of LLMs’ performance in the evaluation process.
>
> W2.a. Both Figures 3.a and 3.b show a set of highly overlapping distributions. The medians of the top contenders in 3.a are barely distinguishable even, however the conclusion that section 3.2 notes is that GPT-4 performs best is mode 1. It seems to me like the authors are making claims without regard to statistical significance of their results.
>
> A2.a. Thank you for your constructive suggestion. Statistical significance can be used to test whether specific features of random variables (e.g., expectation) are different. In our paper, statistical significance is not used for two reasons. First, one indispensable assumption of most hypothesis testing methods (e.g., t-test and chi-square test) is that random variables should follow a family of distributions like normal distribution. Although this assumption is valid in many real-world scenarios (e.g., a person’s height), it is unconfirmed for the random variable of an LLM’s performance, and there is still a lack of research on the statistical property of LLMs’ performance. As a result, we cannot ensure whether the statistical significance itself is correct. Second, rather than to “rank the ability of different LLMs”, we aim to use DynaEval to “discover and depict the ability of different LLMs using evaluation”. Therefore, we presented the difference in LLMs’ performances in terms of average score and stability (the range of the box) in the text of Section 3.2. Following your constructive suggestion, we will discuss more about the performances of LLMs in the next version of our paper.

---

> ### Author Response · Authors · 2023-11-13
> **Reply to Reviewer EKE2 (1-Part 2)**
>
> W2.b. The same is true in Table 2: the difference between GPT-4 (8.77) and ChatGPT (8.74) seems negligible, but that is not reflected in the conclusions of the text.
>
> A2.b. Thank you for your constructive suggestion. As explained in A2.a, we will supplement more discussion about performance of LLMs in Code G&R (especially Table 2) in the next version of our paper.
>
> W3.a. I'm not convinced that DynaEval framework itself brings anything to the table.
>
> A3.a. Thank you for your doubt. As you mentioned, using games as evaluation methods depends on the implementation. One challenge in this process is that games can differ a lot in many aspects such as the number of participants, role design, game rule, etc., and are hard to standardize. As a result, many existing works about LLMs’ performance in games (e.g., [1], [2]) only focus on a specific game and cannot be extended to other games or scenarios. Indeed, one contribution of DynaEval is that it provides a standardized modeling method of all kinds of games using “interaction goals” and “interaction rules” and provides a general framework for running all these games and evaluating the ability of LLMs, as mentioned in A1.b. With this framework, researchers need not waste their time on designing a specific system for running a specific game, but focus more on the properties of the game itself such as what to evaluate (which can be reflected in the interaction goals and interaction rules).
>
> W3.b. The games seem odd, and it's not obvious what capability of the model they are trying to measure. For example, I'm not even sure it makes sense for the Code G&R and the Machine Translation games to actually be turn-based games. As I said, I do see why games can be useful, but the devil is in the details.
>
> A3.b. Thank you for your doubt. As mentioned above, DynaEval is a general framework for running all kinds of games and evaluating LLMs, and “what ability to measure” depends on **the design of games**, which is decided by users of DynaEval. Generally, we can categorize the abilities of LLMs into “fundamental” abilities and “domain-specific abilities”. Fundamental abilities are necessary for all kinds of dynamic interactions between LLMs, such as the ability to understand and reasoning. Domain-specific abilities are required by specific domains or tasks, such as the ability to generate logically correct and well-styled codes in code generation. The design of evaluation tasks can decide the domain-specific abilities to evaluate. For example, as introduced in Section 2.4, p6 in our paper, the Code G&R evaluates the code generation ability of programmer LLMs and the code review ability of reviewer LLMs. In real-world scenarios, users of DynaEval can design their own evaluation tasks to evaluate their required domain-specific abilities of LLMs.
>
> [1] Yuzhuang Xu, Shuo Wang, Peng Li, Fuwen Luo, Xiaolong Wang, Weidong Liu, & Yang Liu. (2023). Exploring Large Language Models for Communication Games: An Empirical Study on Werewolf.
>
> [2] Shenzhi Wang, Chang Liu, Zilong Zheng, Siyuan Qi, Shuo Chen, Qisen Yang, Andrew Zhao, Chaofei Wang, Shĳi Song, & Gao Huang. (2023). Avalon's Game of Thoughts: Battle Against Deception through Recursive Contemplation.

---

> > ### Comment · Reviewer_EKE2 · 2023-11-23
> > **Acknowledging Response - Not convinced**
> >
> > I thank the authors for their response.
> > Unfortunately, I am still not convinced by the responses to my questions. In general, I'll be hard-pressed to find the actual novelty an added value in many aspects of the paper.
> >
> > (W1.b) Agreed, I retract my comment about the interaction goals and rules not being natural. I'm having a hard time putting my thoughts about this into words and that's on my not the authors.
> >
> > (W1.c) I still find equations (1) and (2), as well as the surrounding text, to have very little added information.
> >   -  Not being able to "directly measure" the ability of any models, not just LLMs is a given in the ML field in general. For example, the authors say "... rather than modelling the performance of LLMs by deterministic values, we find that modelling by random variables is more suitable ...". Yes, that's true, but it's also true of all ML models. Even in the classical classification setting, we assume (or hope rather) to have IID samples in the form of a test set. I acknowledge that due to the changing nature of the environment, the sources of randomness are increased, but that doesn't seem to be the point of this section.
> >   - Modelling, usually involves definitions and aims to create usable conclusions, but here the conclusion seems to be "Condition 2" which, if looked at pessimistically, only says that when evaluating LLMs we need more than one samples (N > 1). That's just not good enough.
> >
> > (W1.a) Given the amount of space that I find (in my opinion) being wasted on the previously mentioned content, I'm surprised the authors couldn't spare a few lines or a paragraph to at least summarize the formal definition of EGPI and contrast it with possible other options that they didn't pick.
> >
> > (W2.a) The paper proposes an evaluation method, then presents some results. But when it comes to interpreting the results, with this response, the authors (in my opinion) are just throwing their hands up in the air and saying, sorry since these we are measuring random variables, we cannot compare them in any way.
> >   - First, if that's true, you just admitted to the whole framework being useless. What good is an "evaluation framework" who's results cannot be interpreted? And why are you interpreting them if you just admitted that you cannot?! What good is a "random ranking system"?
> >   - Second, I think the authors could at least make some attempt at rectifying the situation, by actually "modelling" the random variables (which equations (1) and (2) don't do correctly). For example, you could try to find the conditions which would result in each measurement/observation being IID. In that case, you could use the law of large numbers and claim that in aggregate your measurement follows a normal distribution for which a statistical significance test *can* be used.
> >
> > (W3.a and W3.b) I can see the paper having three main contributions: 1) The framework itself, 2) the choice/design of games, 3) the evaluation results performed comparing different models. However, I'm very doubtful of all three aspects of the paper (in the reverse order):
> >   - Evaluations: already expressed doubts about statistical significance, to which I believe you replied by saying that the significance cannot be measured and hence the results are almost useless.
> >   - Choice of games: I believe you said that this is up to the users of DynaEval, hence the games themselves are just stand-ins and of no real value
> >   - Dismissing the framework itself is harder for me, but I'm still not convinced there is anything new or of significance here. I admit that some *works of genius* can seem so obvious after the fact that they can be dismissed unjustly, but I doubt that's what is happening here.

---

### Official Review · Reviewer_U79L · 2023-11-02

**Soundness:** 3 good
**Presentation:** 4 excellent
**Contribution:** 2 fair
**Rating:** 5
**Confidence:** 4

**Summary:**

This paper proposes DynaEval, an interaction-based evaluation framework for evaluating LLMs. DynaEval employs a referee and a message pool in the evaluation process. The authors claim that by this design, the evaluation ensures fairness and stabability. They also discuss the relation between the proposed evaluation procedure and the extensive-form games.

**Strengths:**

1. This paper is very well written and easy to follow. The examples given in the paper help people to understand the procedures of DynaEval.
2. The problem is well motivated. Indeed existing LLM evaluation frameworks rely on high-quality datasets or human judgement, therefore it is necessary to develop novel frameworks to evaluate their ability of interaction.

**Weaknesses:**

Although this paper targets an interesting problem and the presentation is great, I find that the real contribution of this paper is limited or unclear.

First, the proposed DynaEval framework lacks a specific evaluation goal since the "ability of interaction" is too big. DynaEval focuses on the interaction BETWEEN LLMs, whereas I think the "ability of interaction" of LLMs should focus more on the interaction BETWEEN LLMs and HUMAN USERs. In other words, we care more on whether an LLM could understand user intentions and help to execute tasks.Therefore, the ability of interaction should include the ability of understanding, questioning, reasoning or even tool using. I am not sure what specific ability can DynaEval evaluate. The authors may argue that DynaEval is a general framework to evaluate all of these abilities. But I think that is not the core challenge of evaluating LLMs.

Second, I did not find anything novel in DynaEval except the referee. The paper tries to standardize the interaction process and spends many words to emphasize the fairness and stability, and hence propose a synchronous interaction algorithm which ensures anonymity and multiple indenpendent runnings. However, I think most of them are common senses in many evaluation tasks. For example, if we are evaluating several recommendation models, it is very natural to use the same test set (fairness) and run multiple times (stability). Therefore, I did not find much novelty of DynaEval.

Third, the paper tries to build a close relation between DynaEval and extensive-form games with perfect information (EGPI). But I did not see necessity of doing this.  The authors claim that game theory helps to overcome fainess and stability by introducing anonymity and synchronicity. But I think this is unconvincing. Because the concepts of anonymity and synchronicity are not exlusive to game theory. In fact, EGPI can describe a very broad range of interaction scenarios, therefore it is not surprising that the procedures of DynaEval belong to EGPI. But it should be interesting if DynaEval shares some characteristics of EGPI, such as the existence of equilibrium.

**Questions:**

In the example of IDIOMS SOLITAIRE, which specific ability are you testing? From my point of view, winning LLM should have seen more idioms so that it can come up with an answer easily. But does this has anything to do with "interaction"?

---

> ### Author Response · Authors · 2023-11-13
> **Reply to Reviewer U79L (1)**
>
> We would like to thank you very much for the detailed feedback and valuable suggestions.
>
> W1. The proposed DynaEval framework lacks a specific evaluation goal since the "ability of interaction" is too big.
>
> A1. Thank you for the question. We agree with the opinion that LLM evaluation should focus on specific abilities. As for DynaEval, what should be noticed is that **the ability to evaluate depends on the design of the evaluation task, which is determined by the user of DynaEval**. On the other hand, DynaEval itself provides a platform/framework for users to design and implement their evaluation task that evaluates the task-specific abilities of LLMs in dynamic interaction scenarios. For example, in our experiment, the Code G&R evaluates the code generation ability of the programmer LLM and the code review ability of the reviewer LLM. The reviewer may doubt that this task cannot disentangle the understanding and reasoning ability of LLMs because they are also required by this task. However in real-world dynamic interaction scenarios, LLMs always need to use multiple abilities to solve users’ problems, and the understanding and reasoning ability is the basis of them because they need to optimize their solution based on their understanding and reasoning of user feedback. Therefore, as Code G&R and Machine Translation, users of DynaEval can design their own evaluation tasks to evaluate task-specific or domain-specific abilities of LLMs. In our paper, we also introduced in **Appendix A.4** how to design evaluation tasks based on users’ own requirements.
>
> W2. The paper tries to standardize the interaction process and spends many words to emphasize the fairness and stability, and hence propose a synchronous interaction algorithm which ensures anonymity and multiple independent runnings. However, I think most of them are common senses in many evaluation tasks.
>
> A2. Thank you for your question. Indeed, fairness and stability are only easy to achieve in traditional supervised signal-based evaluation or human-based evaluation because the response of each LLM is **independent of and isolated from others**. However, In the dynamic interaction between LLMs, things are much more complicated. LLMs might expose their real identity because of a lack of supervision, and the synchronicity is easy to be violated because the response time of different LLMs can vary a lot. Moreover, the synchronicity also depends on game rules. For example, in the public goods game, LLMs are required to respond simultaneously in each round. But in Code G&R, LLMs are required to respond one by one. In real-world scenarios, the game rule can be much more complex. As a result, to standardize the complex interaction process, we propose the **referee** to help guarantee the fragile but significant anonymity and rate LLMs’ performance, and we propose the **message pool** which uses the synchronous interaction algorithm to compulsively guarantee the synchronicity of the dynamic interaction process.
>
> W3. Third, the paper tries to build a close relation between DynaEval and extensive-form games with perfect information (EGPI). But I did not see necessity of doing this.
>
> A3. Thank you for your question. As mentioned in A2, the dynamic interaction process between LLMs is much more complicated than the traditional supervised signal-based or human-based evaluation process. As a result, we utilize the EGPI to standardize the interaction process in all kinds of tasks and scenarios and introduce the fairness and stability condition of DynaEval. As for shared properties of DynaEval and EGPI such as the existence of equilibrium (Do you mean Nash equilibrium?), since we have proved that DynaEval belongs to EGPI, it is theoretically easy and convenient to analyze the property of DynaEval in terms of game theory. As this paper mainly focuses on LLM evaluation in dynamic scenarios, such an analysis exceeds the range of research of this paper. We plan to analyze the theoretical properties of DynaEval in our next research.
>
> Q1. In the example of IDIOMS SOLITAIRE, which specific ability are you testing? From my point of view, winning LLM should have seen more idioms so that it can come up with an answer easily. But does this has anything to do with "interaction"?
>
> AQ1. As introduced in Section 2.4, p6 in our paper, we test the ability to retrieve and use the Chinese vocabulary of LLMs. Interaction is significant in this task, because LLMs’ responses are based on the context, and Chinese idioms are limited and it is easy for LLMs to acquire a host of idioms. If two participants have seen a similar amount of idioms, they need to use specific strategies to win the game, such as coming up with a Chinese idiom with the last character complex and rare, which makes it hard for the opponent LLM to come up with a valid Chinese idiom, even though it has seen a lot of Chinese idioms.

---

### Official Review · Reviewer_u39q · 2023-11-05

**Soundness:** 3 good
**Presentation:** 3 good
**Contribution:** 3 good
**Rating:** 6
**Confidence:** 3

**Summary:**

The authors in this work propose an evaluation framework for LLMs by connecting the typically interactive use of LLMs to game theory. Leveraging this connection, the authors provide a general framework for evaluating the performance of LLMs as agents in dynamic real-world games. They construct several explicit scenarios from the framework and find relative performance that seems to be somewhat distinct from static evaluation results.

**Strengths:**

The authors make an important point regarding a significant difference in the typical use case versus LLMs and the typical forms of evaluating LLM performance. They provide a simple and theoretically grounded approach to resolving this difference which they explain clearly and soundly. They demonstrate how to use this evaluation criteria with straightforward and well executed experiments. The experimental results seem to demonstrate that they capture a gap between static and dynamic use of LLMs though this result is not explicitly highlighted by the authors - see weaknesses.

**Weaknesses:**

In this work there is a hypothesis that the authors test in their experiment design which they are not explicit about. The hypothesis is that these dynamic evaluation scenarios will uncover different comparative results between LLMs than static evaluation scenarios. It seems to be that they found comparatively different results in their experiments because to my knowledge, static evaluation methods find that ChatGPT underperforms GPT-4 on translation tasks while they find the reverse on their dynamic evaluation dataset. This claim is significant, and should be both highlighted and explained substantially in the text. There needs to be an explicit and clear comparison between the dynamic evaluation results and what static evaluation methods find with an accompanying explanation.

In fact, the authors emphasize an odd point instead of this comparison to the relative performance in static evaluation. They discuss the “improvement of performance” through the evaluation, even while highlighting their main results in the conclusion. While LLMs can improve performance on tasks through interaction, this observation seems completely beside the point of DynEval which is an evaluation procedure, so the absolute performance of the LLMs is not important.

**Questions:**

Is the stability condition reasonable for LLMs? Couldn’t they not converge? Aren’t those sorts of guarantees made for substantially similar agents? Maybe I also don’t correctly understand what “converge” means in this context. Some greater clarity here would help.

The source code link does not work.

---

> ### Author Response · Authors · 2023-11-13
> **Reply to Reviewer u39q (1)**
>
> We would like to thank you very much for the detailed feedback and valuable suggestions.
>
> W1. There needs to be an explicit and clear comparison between the dynamic evaluation results and what static evaluation methods find with an accompanying explanation.
>
> A1. Thank you for your constructive suggestion. Indeed, we conducted the machine translation task on different language pairs. Due to the page limitation, we only put the EN-ZH in the body of the paper and put the rest in Appendix A.9, p20. According to our experiment, ChatGPT only outperforms GPT-4 in the EN-ZH setting of machine translation, while underperforming GPT-4 in the DE-EN and EN-FR setting. Similarly, in the Idioms Solitaire task (which is also relevant to Chinese), we also observed that ChatGPT outperforms GPT-4, as shown in Table 1 and Table 2. These results might demonstrate that ChatGPT can perform better than GPT-4 in Chinese-related tasks. We will discuss more about this discovery in the next version of our paper.
>
> W2. They discuss the “improvement of performance” through the evaluation, even while highlighting their main results in the conclusion. While LLMs can improve performance on tasks through interaction, this observation seems completely beside the point of DynaEval which is an evaluation procedure, so the absolute performance of the LLMs is not important.
>
> A2. We suppose that the “ability to improve performance through interaction” is a significant part of the performance of LLMs. As mentioned in the third paragraph of Section 1, in dynamic interaction scenarios, LLMs repeatedly get feedback and optimize their output to gradually meet users’ requirements. As a result, “how is the performance of an LLM after getting feedback” is significant in our evaluation. In addition, since the feedback highly affects the degree of improvement and is provided by another agent, “how an LLM can improve the performance of another LLM” is also significant in our evaluation.
>
> Q1. Is the stability condition reasonable for LLMs? Couldn’t they not converge? Aren’t those sorts of guarantees made for substantially similar agents?
>
> AQ1. The stability condition is reasonable and indispensable for LLMs. Due to the uncertainty in the text generation of LLM and the variety and complexity of interaction environments (e.g., different interaction contexts can lead to different input for an LLM), the uncertainty of interaction results widely exist in dynamic interaction scenarios and can highly affect the evaluation of LLMs. Therefore, we noticed that the performance of an LLM should be modeled as a random variable rather than a specific value. As a result, we utilize the stability condition of DynaEval to evaluate the expected performance of LLMs, as shown in Eq.(1) in our paper. With multiple running of an evaluation task, the performance of participant LLMs can converge in distribution. This condition is also reasonable for dissimilar agents. As shown in Eq.(2), when modeling the full distribution of an LLM’s performance, the interaction histories (and other LLMs’ abilities) are only viewed as the context and condition of the random variable of the LLM’s performance and can be eliminated using the law of total probability.
>
> Q2. The source code link does not work.
>
> AQ2: Sorry for the confusion. Our source code is available at  https://anonymous.4open.science/r/DynaEval-D334 . We will update the source code link to the new version of our paper.

---

> ### Comment · Reviewer_u39q · 2023-11-21
> **Reply to rebuttal**
>
> Thank you for the responses.
>
> However, I have decreased my score with the rebuttal because there is definitely confusion about what this relative performance evaluation is compared to static evaluation. When you clarified this in the first rebuttal comment, you still are replied to total score between 2 LLMs only in the dynamic setting. I still am in support of this paper, but for it to be a meaningful benchmark, it needs to clearly find that comparing 2 LLMs (i.e. ChatGPT and GPT-4) yields different results in this test than the existing ones already defined in prior work.
>
> It is critical that you very definitively point out "on [this task (for example machine translation)], static evaluation methods yield [this relative performance] between 2 LLMs and our dynamic evaluation method uncovers [this relative performance]". Please clearly write this for all tasks for which the relative performance is not the same as a comment (ultimately to be included in the paper) for me to fully support it in further discussion.

---

### Official Review · Reviewer_ZvMp · 2023-11-08

**Soundness:** 1 poor
**Presentation:** 2 fair
**Contribution:** 1 poor
**Rating:** 3
**Confidence:** 4

**Summary:**

This paper proposes a dynamic interaction-based evaluation framework for evaluating LLMs named DynaEval. Inspired by game theory, the authors highlight the interaction process of LLMs is theoretically aligned with extensive games with perfect information (EGPI). The interaction process of LLMs in DynaEval is categorized into four steps: selection, interaction, recording, and circulation. The authors propose a fairness condition as letting LLMs be **anonymous** and delivering the messages **synchronously**. The authors specify a stability condition as running multiple times of interactive LLMs independently. In practice, the authors introduce a referee and message pool to implement the fairness and stability condition.

**Strengths:**

- The evaluation of LLMs with fairness and stability guarantee is a less studied yet very important research problem for the machine learning community. The topic of this paper is highly related to the ICLR conference.

- The paper is well-written and easy to read.

- The intuition of relating the interaction process of LLMs to the extensive games with perfect information (EGPI) is novel, though I feel the connection fails to result in a strong technical contribution.

**Weaknesses:**

- Overall, I feel the technical contribution of this paper is limited because the method fails to come up with a practical and novel fairness condition or stability condition. The presented fairness condition and stability condition are both standard practice, with limited new insights. Anonymity, synchronicity, and independent runs are all typical practices during LLM evaluation. As the method is tackling dynamic interaction of LLMs, I do not see any specific part of the algorithm being customized to the specific property of dynamic interaction. Also, I feel the concepts of referee and message pool are not novel either.

- Apart from the refee and message pool, another important concept that is less talked about in the paper is the **task rule**. The task rule itself has a lot to do with fairness and stability for the evaluation task. Regretfully, in the paper, very little details about it has been revealed.

- The referee plays a very important role in the evaluation, responsible for the selection, recording, and circulation process.  The referee also takes charge of evaluating all LLMs' performance. However, for the fairness evaluation of LLMs, it is unclear whether it is possible to develop referees automatically in a supervised signal-based manner with less human-based guidance.

- The discussion on the proposed evaluation method compared to existing ones is quite weak. It is unclear how the proposed one can tackle the dynamic nature of the LLMs interaction while the existing evaluation methods cannot.

**Questions:**

Please refer to the WEAKNESSES part.

---

> ### Author Response · Authors · 2023-11-13
> **Reply to Reviewer ZvMp (1-Part 1)**
>
> We would like to thank you very much for the detailed feedback and valuable suggestions.
>
> W1. The technical contribution of this paper is limited because the method fails to come up with a practical and novel fairness condition or stability condition. Also, the concepts of referee and message pool are not novel either.
>
> A1. Thank you for your doubt. The technical contribution of this paper is that: 1. We noticed and proved the relationship between the dynamic interaction process of LLMs in real-world scenarios and EGPI; 2. Based on EGPI, we proposed DynaEval which is a **general** framework for dynamic LLM evaluation that can cover most interaction processes in real-world scenarios, which is much more complex than traditional signal-based evaluation methods. Anonymity, synchronicity, and independent runs are fundamental yet indispensable in all dynamic interaction-based LLM evaluation. However, they are only easy to achieve in traditional supervised signal-based evaluation or human-based evaluation because **the response of each LLM is independent of and isolated from others**. In the dynamic interaction between LLMs, things are much more complicated. **LLMs might expose their real identity because of a lack of supervision, and the synchronicity is easy to be violated because the response time of different LLMs can vary a lot. Moreover, the synchronicity also depends on game rules.** For example, in the public goods game, LLMs are required to respond simultaneously in each round. But in Code G&R, LLMs are required to respond one by one. In real-world scenarios, the game rule can be much more complex. As a result, we propose the **referee** to help guarantee the fragile but significant anonymity and rate LLMs’ performance, and we propose the **message pool** which uses the synchronous interaction algorithm to compulsively guarantee the synchronicity of the dynamic interaction process.
>
> W2. The task rule itself has a lot to do with fairness and stability for the evaluation task, but very little details about it has been revealed.
>
> A2. Thank you for your question. As illustrated in Section 2.2 and Definition 1, the task rule of a dynamic interaction-based evaluation task consists of an **interaction goal** and an **interaction rule**. Such a design is simple yet sufficient for depicting all dynamic interaction-based tasks in real-world scenarios.
>
> As for fairness and stability, they are disentangled from task rules, and achieved by the special mechanism of our proposed DynaEval (referee, message pool, etc.) that is irrelevant to specific task. This design is in alignment with real-world requirements. For example, in the review-rebuttal process of this paper, the review-rebuttal rule only regularizes the interaction goal and the interaction rule for reviewers and authors. For reviewers, the goal is to accept high-quality papers and reject low-quality papers while helping improve the quality of the latter. For authors, the goal is to make their paper accepted by the conference and improve its quality. The interaction rule is that reviewers first review the paper and give the comment. Then authors and reviewers can discuss the paper during the rebuttal phase. During this process, stability is ensured by assigning multiple reviewers to a paper, and fairness is ensured by the surveillance of the OpenReview platform. Indeed, even though the task rule does not regularize fairness and stability, they can still be satisfied by the special mechanism of DynaEval.

---

> ### Author Response · Authors · 2023-11-13
> **Reply to Reviewer ZvMp (1-Part 2)**
>
> W3. For the fairness evaluation of LLMs, it is unclear whether it is possible to develop referees automatically in a supervised signal-based manner with less human-based guidance.
>
> A3. First, it is hard to develop referees automatically in a supervised signal-based manner in DynaEval because the interaction environment is complex, and the interaction rule and evaluation metric of different dynamic interaction-based evaluation tasks can vary a lot in real-world scenarios. As a result, it is inefficient to customize a supervised signal-based referee for each (countless) evaluation task. Moreover, during the interaction process, participant LLMs may generate abnormal outputs (such as outputs that violate the format) which affect the running of the evaluation process and should be distinguished by the referee. However, supervised signal-based referees cannot distinguish these abnormal outputs. Therefore, supervised signal-based referees are hard to develop in DynaEval.
>
> Second, human-based guidance for referees does not affect the fairness of LLM evaluation. As illustrated in Condition 1 in our paper, all participant LLMs should be anonymous during the interaction process. This condition means that participants are not only anonymous to each other, but also anonymous to the referee. That is, the referee can only identify each participant by their assigned ID. As a result, the referee can only evaluate the performance of LLMs based on their interaction history, which is identity-irrelevant. As a result, the fairness of LLM evaluation can still be achieved when using referees with human-based guidance.
>
> W4. The discussion on the proposed evaluation method compared to existing ones is quite weak. It is unclear how the proposed one can tackle the dynamic nature of the LLMs interaction while the existing evaluation methods cannot.
>
> A4. As presented in **the third paragraph of Section 1**, we discussed the reason why existing evaluation methods cannot tackle the dynamic nature of the LLMs interaction. In the fourth paragraph, we further illustrated how the proposed DynaEval can solve this problem. For traditional supervised signal-based methods, their evaluation environments are static, and hard to capture the complexity of dynamic real-world situations. Moreover, although some evaluation methods support multiple-round conversations, they still fall short in providing a dynamic interaction environment and lack generality.

---

### Public Comment · ~Guohao_Li1 · 2023-11-14
**Suggesting related work**

The paper introduces DynaEval, a dynamic interaction-based framework for evaluating Large Language Models (LLMs) in real-world scenarios. Traditional LLM evaluation methods, mostly static and domain-specific, are inadequate for dynamic, multi-agent environments where ongoing feedback and interactions shape the outcomes. DynaEval, inspired by game theory, addresses this gap by equating interaction processes with dynamic games, ensuring fairness and stability in evaluations. Key components include a referee for supervising interactions and standardizing outputs, and a message pool for managing interaction histories. Four tasks - Public Goods Game, Idioms Solitaire, Code Generation and Review, and Machine Translation - were used to test DynaEval, with models like ChatGPT, GPT-4, Claude-2, and PaLM evaluated. The results demonstrated DynaEval's effectiveness in fair and stable assessment of LLMs, with dynamic interactions enhancing output quality.

Thanks for the great work. It could also be beneficial to discuss prior work on multi-LLM agents for the study of cooperative interactions [1].

[1] Li, Guohao, Hasan Abed Al Kader Hammoud, Hani Itani, Dmitrii Khizbullin, and Bernard Ghanem. "CAMEL: Communicative Agents for" Mind" Exploration of Large Language Model Society." NeurIPS 2023

---

> ### Author Response · Authors · 2023-11-15
> **Reply to Public Comment**
>
> Thanks for your approval to our work. We will discuss the work on multi-LLM agents for the study of cooperative interactions that you mentioned above and follow more similar prior works in the future.

---

### Author Response · Authors · 2023-11-17
**[Paper # 5534] Have our responses resolved your questions?**

Dear reviewers:

Thank you very much for your insightful reviews and constructive suggestions for our work. We have posted our response and tried our best to resolve your questions and doubts. Have our responses resolved your questions and doubts? Your feedback is significant to us.

Thank you again for your time and patience!

Sincerely,

Authors of Paper # 5534

---

### Meta-Review · Area_Chair_N85q · 2023-12-09

**Metareview:**

This proposes DynaEval -- a game-based evaluation framework, loosely motivated by EGPI, and evaluate some avaiable LLMs on these tasks. I think the idea of evaluating LLMs on games and long-form multi-turn interactive tasks is interesting and well-accepted, but it is unclear if this paper makes a signfiicant contribution along these lines. For instance, the paper evaluates multiple models,  but in discussion with a reviewer, the authors say that statistical significance cannot be judged exactly. Second, and perhaps more important, it is unclear / less analyzed what these precise choice of tasks bring to the table for LLM evaluation and development over other kinds of games where the environment is modelled by another LLM. I think Reviewer EKE2's discussion contains most of my concerns as well, and I would encourage authors to carefully take the reviewer comments into account for revising the paper.

**Justification For Why Not Higher Score:**

Mostly lack of clarity of the contribution and signfiicance compared to other evaluations

**Justification For Why Not Lower Score:**

N/A

---

### Decision · Program_Chairs · 2024-01-16

Reject